# DFSSATTEN: DYNAMIC FINE-GRAINED STRUCTURED SPARSE ATTENTION MECHANISM

## ABSTRACT

Transformers are becoming mainstream solutions for various tasks like NLP and Computer vision. Despite their success, the quadratic complexity of their attention mechanism hinders them from applying to latency-sensitive tasks. Tremendous efforts have been made to alleviate this problem, and many of them successfully reduce the asymptotic complexity to linear. Nevertheless, few of them achieve practical speedup over the original full attention, especially under the moderate sequence length. In this paper, we present DFSSATTEN, an attention mechanism that dynamically prunes the full attention weight matrix to the 50% fine-grained structured sparse pattern used by the sparse tensor core on NVIDIA A100 GPU. We provide both theoretical and empirical evidence that demonstrate DFSSATTEN is a good approximation of the full attention mechanism and can achieve speedups in wall-clock time under arbitrary sequence length. We evaluate our method on tasks from various domains under different sequence lengths from 256 to 4096. DFSSATTEN achieves $1.27 \sim 1.89\times$ speedups over the full-attention mechanism with no accuracy loss on A100 GPU.

## 1 INTRODUCTION

Transformers (Vaswani et al., 2017) have achieved competitive performance across various domains like NLP (Ott et al., 2018) and Computer Vision (Dosovitskiy et al., 2021). The key feature that sets them apart from traditional neural network architectures is the attention mechanism (Vaswani et al., 2017), which allows the transformers to gather information from the embeddings of elements in the input sequence in an adaptive and learnable manner.

Nevertheless, the high computation cost and memory footprint brought by the attention mechanism make it difficult to apply transformers to latency-sensitive tasks. Many efficient attention mechanisms(Tay et al., 2020b; Zaheer et al., 2020; Beltagy et al., 2020; Tay et al., 2020a; Roy et al., 2021; Kitaev et al., 2020) have been proposed to address this issue. However, most of them drastically modify the original full attention mechanism and introduce a handful of hyper-parameters to tune. Therefore, they require tremendous engineering effort to deploy and optimize. Besides, they usually need to be trained from scratch instead of exploiting pretrained models like BERT (Devlin et al., 2019). Some of them rely on fixed sparse patterns or extremely high sparsity to achieve wall-clock time speedup. Therefore, these methods usually require thousands of pretraining or fine-tuning steps on specific tasks and toilsome tuning of several hyper-parameters to reach good accuracy. Last but not least, previous methods usually introduce additional operators like top-k, sort that cause large overheads and offset their benefits at moderate sequence length.

In this paper, we present DFSSATTEN, a simple and effective sparse attention mechanism that address the limitations mentioned above. DFSSATTEN *dynamically* prunes the full attention score matrix using 50% fine-grained structured sparse patterns (NVIDIA, 2020). This pattern is GPU friendly and can leverage the new sparse tensor core on NVIDIA A100 GPU (Mishra et al., 2021). Our DFSSATTEN offers several advantages over existing studies:

- It requires minimal changes to the original full-attention with no hyper-parameters to tune. This makes it a drop-in replacement of the full attention that only requires to change a few lines of code. Moreover, it can directly exploit existing pretrained models like BERT (Devlin et al., 2019) and RoBERTa (Liu et al., 2020).

- We dynamically prune the attention scores based on their magnitude under only 50% sparsity. This allows the pruned attention matrix to reserve the important entries, achieving on par model accuracy with full attention even without fine-tuning.

- Our method introduces zero overhead on existing GPU hardware. As a result, we are able to achieve wall-clock time speedup and memory footprint reduction over the full attention in arbitrary sequence length.

To conclude, our main **contributions** are summarized below:

- We propose DFSSATTEN, a dynamic sparse attention mechanism that is a drop-in proxy of the full attention mechanism. Its effectiveness is justified by both empirical and theoretical evidence.

- We present a dedicated CUDA kernel design to completely remove the pruning overhead. The pruning is implemented as an epilogue of the dense matrix multiplication which produces the attention score matrix.

- We evaluate DFSSATTEN on tasks cross various domains and sequence lengths. It achieves $1.27 \sim 1.89\times$ speedup over the full attention with no accuracy loss.

## 2 BACKGROUND AND MOTIVATION

We first introduce the preliminaries, notations, and background of our paper.

### 2.1 FULL ATTENTION MECHANISM

Given an input sequence $\boldsymbol{X} = (\boldsymbol{x}_1, .., \boldsymbol{x}_n) \in \mathbb{R}^{n \times d}$, the full attention mechanism can be defined as

$$\boldsymbol{O} = Softmax(\boldsymbol{Q}\boldsymbol{K}^T/\sqrt{d})\boldsymbol{V}, \tag{1}$$

where $\boldsymbol{Q} = \boldsymbol{X}\boldsymbol{W}_q$, $\boldsymbol{K} = \boldsymbol{X}\boldsymbol{W}_k$, and $\boldsymbol{V} = \boldsymbol{X}\boldsymbol{W}_v$ are query, key, and value matrices. $\boldsymbol{Q}\boldsymbol{K}^T$ forms a full-quadratic adjacency matrix, whose edge weights are the dot-product similarity between all the elements in the sequence. This adjacency matrix is standardized with $1/\sqrt{d}$ to keep the unit second moment and then normalized with softmax. At last, the row feature vectors in $\boldsymbol{V}$ are aggregated according to the normalized adjacency matrix by multiplying them together. In the rest of this paper, we denote $\boldsymbol{A} = Softmax(\boldsymbol{Q}\boldsymbol{K}^T/\sqrt{d})$ for simplicity. We refer $\boldsymbol{Q}\boldsymbol{K}^T$ as the attention score matrix and $\boldsymbol{A}$ as the attention weight matrix.

### 2.2 EFFICIENT ATTENTION MECHANISM

The high computation cost and memory footprint in the full attention mechanism come from $\boldsymbol{A}$, whose size grows quadratically with the sequence length $n$. To address this issue, various efficient attention mechanisms have been proposed (Tay et al., 2020b).

**Fixed Sparse Patterns**. Zaheer et al. (2020); Beltagy et al. (2020) apply a set of fixed sparse attention patterns on $\boldsymbol{A}$, like global attention and sliding window attention. These patterns are constructed from empirical observations and designed GPU-friendly to achieve wall-time speedup. However, as these patterns are designed empirically and fixed during inference, there is no guarantee that they can always capture the important entries in $\boldsymbol{A}$ or transfer easily across different tasks.

**Dynamic Sparse Patterns**. Ham et al. (2021) dynamically generate fine-grained sparse attention patterns on $\boldsymbol{A}$ with low-cost binary hashing. However, this technique requires specialized hardware to achieve speedup, so it is not available on general-purpose hardware like GPU. Tay et al. (2020a); Roy et al. (2021); Kitaev et al. (2020) apply various clustering methods and only compute the attention within each cluster. Although computing full attention in each cluster is more friendly to GPU compared with fine-grained sparsity, the clustering methods contain several GPU-unfriendly operators like top-k and sorting that offsets their benefits under moderate sequence length.

**Low Rank / Kernel**. Wang et al. (2020) project $\boldsymbol{A}$ from $n \times n$ to $n \times k$ with linear projection. Choromanski et al. (2021) introduce the FAVOR+ which approximates the softmax with kernel method. This allows them to change the computation order and reduce the asymptotic complexity

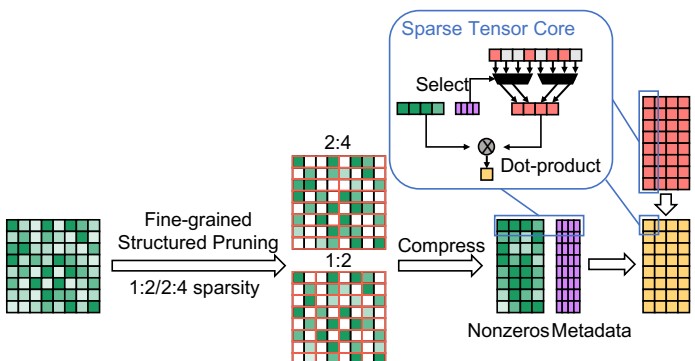

Figure 1: A100 GPU Fine-Grained Structured Sparsity Pruning. (NVIDIA, 2020)

to linear. However, the low-rank projection and kernel construction also introduce considerable overhead. This makes these methods only effective under long sequence length.

Besides, the previous studies drastically change the attention mechanisms, tens of thousands pre-training or finetuning steps are required to reach a comparable performance with the origin full attention mechanism. So they require tremendous engineering effort to deploy.

### 2.3 FINE-GRAINED STRUCTURED SPARSITY IN NVIDIA A100 GPU

NVIDIA introduces the fine-grained structured sparsity in the A100 GPU. As shown in Figure 1, the dense input matrix is pruned with fine-grained structured pruning. If the data type is float, 1:2 sparsity is used which selects the larger one in two consecutive entries. If the data type is bfloat16 or float16, the 2:4 sparsity is used which selects two larger ones among four consecutive elements. After the pruning, the result is compressed to nonzeros and metadata. The nonzeros contain the value of reserved data that is 50% smaller than the original one. The metadata records the index of the nonzeros in the origins matrix. It takes 4 bit metadata to record the decision of each 1:2 or 2:4 selection. Therefore, the metadata is only 1/16 of the original dense matrix in terms of bits. This compressed sparse matrix can be multiplied with a dense matrix under the support of the sparse tensor core to achieve significant speedup.

This fine-grained structured sparsity has been applied to the static weight matrices in various neural network models including transformer (Mishra et al., 2021). It can effectively accelerate the feed-forward part of the transformer up to $1.9\times$. However, to the best of our knowledge, no previous studies use it in the attention mechanism where the attention weight matrix is *dynamically* generated for each sequence. One plausible explanation is that during pruning, GPU must read the whole dense matrix to be pruned from the memory. Then, after selecting the elements to be reserved under the 1:2 and 2:4 pattern, it must also generate the metadata encoded in a special format such that the metadata can be used efficiently later. All these overheads will offset the benefit brought by the pruning if we do it on the fly.

## 3 DYNAMIC FINE-GRAINED STRUCTURED SPARSE ATTENTION MECHANISM

In this section, we first give an overview of our DFSSATTEN method. Then, we discuss the design considerations of exploring sparsity in attention and the choice of sparse granularity in our method for GPU-friendly implementation and effectiveness. Finally, we briefly introduce our GPU kernel design to remove pruning overhead.

Our proposed DFSSATTEN mechanism is simple and effective, as illustrated in Figure 2. Compared with the full-quadratic attention mechanism, our method dynamically prunes attention scores without incurring storage or computation overhead, while maintaining the effectiveness of attention. More importantly, our method can achieve practical speedups of attention on existing GPU hardware with customized CUDA kernels. Listing 1 shows all the modifications to be made to use DFSSATTEN.

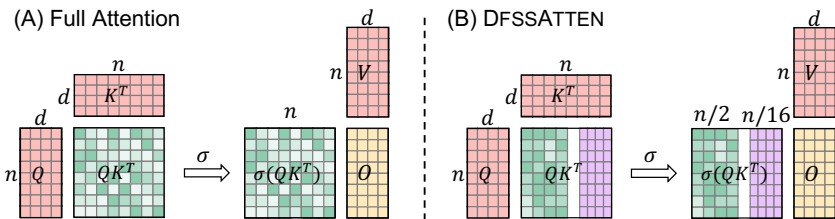

Figure 2: Overview of our Dynamic Fine-grained Structured Sparse Attention Mechanism. $\sigma$ represents *softmax*, $\frac{1}{\sqrt{d}}$ is omitted for simplicity.

```python
# Full attention mechanism
def full_attention(q,k,v):
    attn_weight = torch.bmm(q, k.transpose(1, 2))
    attn_weight = torch.nn.functional.softmax(attn_weight, -1)
    return torch.bmm(attn_weight, v)

# DFSS attention mechanism
import dspattn
def dfss_attention(q,k,v):
    attn_weight, metadata = dspattn.bmm(q, k)
    attn_weight = torch.nn.functional.softmax(attn_weight, -1)
    return dspattn.spmm(attn_weight, metadata, v)
```

Listing 1: Example of using DFSSATTEN. The "dspattn" is the package we developed.

## 3.1 DESIGN CONSIDERATIONS FOR EXPLOITING ATTENTION SPARSITY

As illustrated in Figure 3, the attention mechanism can be considered as three stages: $QK^T$, $Softmax$, and $AV$. To design a sparse attention mechanism, the first decision to make is where should we induce pruning to sparsify the attention.

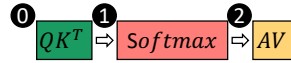

Figure 3: Attention Stages.

If we start from ❶, all the three stages will be benefited from the sparsity given effective implementation on GPU: the dense matrix multiplication between $Q$ and $K$ will be replaced with the sampled dense-dense matrix multiplication (SDDMM) which only computes the entries identified by the sparse pattern. The $Softmax$ only operates on the nonzero values in each row. The original dense matrix multiplication between $A$ and $V$ will be replaced with a sparse matrix-matrix multiplication (SpMM) which multiplies a sparse matrix with a dense matrix. However, as it is not possible to exactly know which entry in $QK^T$ has higher magnitude before computing $QK^T$, starting from ❶ usually requires some additional components to predict the location of important entries.

Starting from ❶ requires us to compute a dense matrix multiplication between $Q$ and $K$. The benefit is that we can explicitly select important entries from $QK^T$ without prediction. As the softmax is a monotonically increasing function, starting from ❷ does not offer any benefits over ❶ but throws away the opportunity to accelerate $Softmax$.

In this paper, we choose to start from ❶ based on two considerations. First, replacing the dense matrix multiplication with SDDMM at $QK^T$ offers limited speedup even at high sparsity. Chen et al. (2021b) show that it is difficult for SDDMM to achieve speedup over its dense counterpart under 80% sparsity even with some structured design. Second, starting from ❶ allows us to keep our design simple such that it does not introduce additional overhead or hyper-parameters to tune.

## 3.2 ON THE GRANULARITY OF SPARSE ATTENTION PATTERNS

The second decision to make is what sparse pattern to use as it will tremendously affect the latency of SpMM as well as the overhead to encode the sparse $QK^T$. Existing studies exploit various sparse encoding schemes. For instance, the compressed sparse row (CSR) is popular for encoding fine-grained sparsity. However, CSR-based SpMM requires over 95% sparsity to be on par with its dense counterpart (Chen et al., 2021b). Block sparsity is also widely used as it can bring considerable wall-time speedup at moderate sparsity given large block size. However, it cannot capture some

fine-grained attention patterns. Moreover, these patterns require data comparisons within the entire row, which is difficult to execute in parallel and unfriendly to GPUs.

We find the fine-grained structured sparsity mentioned in Section 2.3 is a good choice as long as we address the pruning overhead. On one hand, the 1:2 and 2:4 selections are performed locally and are easy to execute in parallel. On the other hand, the size of compressed nonzeros is half of the original dense attention matrix, so the softmax only needs half of the computations. Powered by the NVIDIA Sparse Tensor Core, the SpMM between the compressed $A$ and $V$ can also achieve $1.7\times$ speedup.

### 3.3 Empirical Results of DfssAtten Mechanism

Empirically, we find this pattern can well approximate the full attention mechanism. We first finetune a BERT-large model on SQuAD

Table 1: F1 Score w/o Finetune on SQuAD v1.1

| Full | 1:2 | 2:4 |
|------|-----|-----|
| $93.17 \pm 0.27$ | $92.86 \pm 0.22$ | $93.00 \pm 0.16$ |

v1.1 under full attention. Then, we directly replace the full attention with the 1:2 and 2:4 attention without additional finetuning. The F1-scores are summarized in Table 1 under $Cl = 95\%$. It is obvious that the accuracy loss is only around one sigma even without finetuning.

### 3.4 Removing Pruning Overhead

As mentioned in Section 2.3, the major challenge that hinders us from using the fine-grained structured sparse attention is the pruning overhead. We observe that when computing $QK^T$, the results are first accumulated in GPU registers and written to memory when all the computations are done. Therefore, we can implement the pruning as an epilogue of the matrix multiplication: after the accumulation is finished, we compare the data stored in the registers, select the larger ones and generate the metadata. Then, we only write the reserved non-zeros and metadata to memory. This design brings two benefits. First, it completely removes the overhead caused by reading the matrix to be pruned from memory, so it has zero overhead. Second, the memory footprint caused by the attention weight matrix is reduced from $n^2 \times 32\text{-}bit$ to $\frac{n^2}{2} \times 32\text{-}bit + \frac{n^2}{16} \times 32\text{-}bit$ as the original $n \times n$ full attention weight matrix is not written to memory. The more detailed description of the CUDA kernel design including how to encode the metadata on the fly is summarized in Appendix A.1.

## 4 Theoretical Results

In this section, we provide more theoretical and empirical evidence that justify our DfssAttn as a good proxy of the full attention mechanism. The strategy is to first derive the theoretical value of 1) quality of the approximation with different sparse patterns 2) speedup can be achieved under certain sparsity. Then, we compare the quality of different methods under the same speedup.

### 4.1 Attention Lottery Ticket

We borrow the lottery ticket hypothesis (Frankle & Carbin, 2019) and extend it to the attention mechanism. The last step $AV$ in the attention mechanism can be viewed as the aggregation in the graph neural network. Following the Generalized Attention Mechanism (Zaheer et al., 2020), we describe it with a weighted directed graph $\mathcal{G} = (A, X)$. $A$ is the adjacent matrix and $A_{u,v} > 0$ indicates that element $x_u$ attends to $x_v$. Inspired by the *Graph Lottery Tickets* (Chen et al., 2021a), we propose the *Attention Lottery Ticket* as follows.

**Attention Lottery Ticket (ALT)**. Given a fully connected d graph $\mathcal{G} = \{A, X\}$ constructed from the full quadratic attention mechanism (Vaswani et al., 2017), the associated sub-graph can be defined as $\mathcal{G}_s = \{m \odot A, X\}$, where $m$ is a binary mask. If a $\mathcal{G}_s$ has the performance matching or surpassing the original full quadratic attention mechanism, then we define the sparse attention mechanism with $\mathcal{G}_s$ as an *attention lottery ticket*.

Zaheer et al. (2020) have proved the existence of lottery tickets by showing 1) sparse attention mechanisms are universal approximators of sequence to sequence functions when being used as encoder 2) sparse encoder-decoder transformers are Turing Complete. So the remaining problem is how to identify the winning tickets $\mathcal{G}_s$ at runtime.

## 4.2 QUALITY OF THE LOTTERY TICKET

A popular strategy that empirically works well is selecting the top-k neighborhood in $\mathcal{G}$ based on the magnitude of edge weight. We refer it as *Top-k Sparsity*. Intuitively, this strategy is based on the hypothesis that the edges with larger edge weight are more important. It has been widely adapted in existing studies (Frankle & Carbin, 2019; Chen et al., 2021a; Ham et al., 2021; Wang et al., 2021) and demonstrated its ability to preserve model accuracy at a high sparsity ratio. Following this trend of work, we define the Quality of Attention Lottery Ticket as follows:

**Definition 4.1.** *($L^p$-Quality of Attention Lottery Ticket) The quality of attention lottery ticket $\mathcal{G}_s = \{m \odot A, X\}$ under density $s = \frac{1}{n^2} \sum_{j=1}^{n} \sum_{i=1}^{n} m_{j,i}$ is defined as*

$$\mathcal{Q}^p = \frac{1}{n} \sum_{j=1}^{n} \frac{\sum_{i=1}^{n} (m \odot A)_{j,i}^p}{\sum_{i=1}^{n} A_{j,i}^p}. \tag{2}$$

The above definition computes the expectation of normalized $L^p$ norm in each row of the attention score matrix. The $p$ is a task-dependent factor that indicates how the accuracy depends on the edges with higher magnitude. In this paper, we compare the $L^p$-Quality of tickets yield by three types of sparse patterns: Top-K, fixed, and our dynamic 1:2 and 2:4 sparse pattern. Particularly, we have the proposition below:

**Proposition 4.1.** *Under the assumption that the entries in $QK^T/\sqrt{d}$ follow i.i.d. $\mathcal{N}(\mu, \sigma)$, we have*

$$\mathcal{Q}_{topk}^p|_s \approx \frac{1 + erf\left(\frac{p\sigma}{\sqrt{2}} - erfinv(1 - 2s)\right)}{2}, \quad \mathcal{Q}_{fix}^p|_s = s, \quad \mathcal{Q}_{2:4}^p \geq \mathcal{Q}_{1:2}^p = \frac{1 + erf\left(\frac{p\sigma}{2}\right)}{2} \tag{3}$$

*(Proof: Appendix A.2)*

It is obvious that the $\mathcal{Q}_{topk}^p$ achieves the upper bound of $Q^p$ under $s$. Besides, the $p\sigma$ is always positive. Therefore, we also have $\mathcal{Q}_{2:4}^p \geq \mathcal{Q}_{1:2}^p > \mathcal{Q}_{fix}^p|_{s=0.5} = 1/2$.

## 4.3 EFFICIENCY OF THE LOTTERY TICKET

A lottery ticket with high quality does not necessarily mean that it is also efficient to execute for wall-clock time speedup. In this section, we analyze the efficiency of the three sparse patterns.

**Top-K Sparsity**. Zhao et al. (2019) explicitly select k neighbors in each row of $A$ based on their magnitude. However, as shown in their Table 4, the explicit sparse transformer has lower inference throughput despite $k \ll n$. On one hand, the top-k operator is difficult to parallel and introduces high overhead. On the other hand, even if an oracle top-k sparsity mask $m$ were provided with zero overhead, it would still be difficult for the explicit Top-K sparse attention to beat its dense counterpart. We provide a theoretical upper bound for density $s$ in Proposition 4.2.

**Proposition 4.2.** *Given embedding size $d$ and the maximum tiling size supported by GPU $T$, the upper bound of the speedup achieved by Top-K Sparsity under density $s$ is (Proof: Appendix A.3)*

$$Speedup < \frac{4d + 3T}{2d + T + (d + 2T + dT)s}. \tag{4}$$

As typical values for the dimension $d$ and tiling size $T$ are $d = 64, T = 128$, $s < 4.5\%$ is a necessary and insufficient condition to have $Speedup > 1$. Notably, this is not a strict upper bound as we did not take the overhead of identifying top-k entries into consideration. Therefore, the strict upper bounder should be even smaller.

**Fixed Sparsity**. As the fixed sparse pattern are designed or learned before inference, they can be designed to be GPU-friendly and have the same tiling size with the dense matrix multiplication. Therefore, we can derive the upper bound of the speedup under density $s$ with the same strategy in Proposition 4.2:

$$Speedup = \frac{n^2\left(\frac{2d}{T} + 1\right) + 2n^2 + nd\left(\frac{2n}{T} + 1\right)}{sn^2\left(\frac{2d}{T} + 1\right) + 2n^2s + nd\left(\frac{(1+s)n}{T} + 1\right)} \overset{n \gg d}{=\!=\!=} \frac{4d + 3T}{(1 + 3s)d + 3sT}. \tag{5}$$

**Dynamic 1:2 / 2:4 Sparsity**. Similarly, we can derive the theoretical speedup with 1:2 and 2:4 sparsity as follows

$$Speedup = \frac{n^2\left(\frac{2d}{T}+1\right)+2n^2+nd\left(\frac{2n}{T}+1\right)}{n^2\left(\frac{2d}{T}+\frac{1}{2}+\frac{1}{16}\right)+n^2+nd\left(\frac{n}{T}+\frac{n}{2T}+\frac{n}{16T}+1\right)} \overset{n\gg d}{\gtreqqless} \frac{64d+48T}{57d+25T}. \quad (6)$$

### 4.4    QUALITY OF THE LOTTERY TICKETS UNDER THE SAME EFFICIENCY

With the theoretical conclusions above, we compare the quality of the lottery ticket under our dynamic 1:2 and 2:4 sparsity with the other two methods under the same efficiency.

**Comparison with Top-K Sparsity**. The Top-K sparsity achieves the same efficiency with ours at

$$s < \frac{(4d+3T)(57d+25T)}{(64d+48T)(d+2T+dT)} - \frac{2d+T}{(d+2T+dT)}. \quad (7)$$

With typical values $T=128, d=64$, we have $s<0.02$. We can substitute it to Proposition 4.1 and get $\mathcal{Q}_{topk}^p < \mathcal{Q}_{1:2}^p$ when $p\sigma < 7$. On the other hand, when $p\sigma > 7$, although the Top-K sparsity produces tickets with higher quality, $\mathcal{Q}_{1:2}^p|_{p\sigma=7} \approx 0.9999996$ is already very close to 1.

**Comparison with Fixed Sparsity**. The fixed sparsity achieves the same efficiency with ours when

$$s = \frac{(4d+3T)(64d+48T)}{(57d+25T)(3d+3T)} - \frac{d}{3d+3T}. \quad (8)$$

With typical values $T=128, d=64$, we have $s\approx 0.63$. On the other hand, we have theoretical value of $\sigma \approx 1$ and $p \geq 1$ . The $p \geq 1$ is based on the observation that the edges with higher magnitude are more influential. Therefore, we have $p\sigma \geq 1$ and $\mathcal{Q}_{1:2}^p \geq 0.76 > 0.63 = \mathcal{Q}_{fix}^p|_{s=0.63}$.

To conclude, compared with both top-k sparsity and fixed sparsity, our method can always yield lottery tickets with higher quality under the same efficiency. To support this conclusion, we further provide some empirical studies in Appendix A.4. Besides, we found both theoretically and empirically that our method is a good complementary to the kernel-based transformers like Performer (Choromanski et al., 2021). We add more discussions about it in Appendix A.5.

## 5    EVALUATION

In this section, we first evaluate the accuracy of our dynamic fine-grained structured sparse attention mechanism on tasks across different domains. Then, we profile our methods on NVIDIA A100 GPU under different sequence lengths from 256 to 4096 to show that we can achieve practical speedup in arbitrary sequence length.

### 5.1    MODEL ACCURACY

To show that our method is effective in comprehensive scenarios, we first evaluate the model accuracy on tasks in different domains and sequence length. For models under "bfloat16" data type, we first finetune them from the pretrained model under "float" data type as "float" provides more precise gradient that helps convergence. After the finetuning, we directly cast all the parameters in the model to "bfloat16" and test it on the test dataset. For Question Answering and Masked Language Modeling tasks, we report the results averaged over 8 runs under different random seeds.

**Question Answering**. We evaluate the BERT-large on SQuAD v1.1 under sequence length 384. We use the "bert-large-uncased-whole-word-masking" in Huggingface (Wolf et al., 2020) as the pretrained model and finetune it with the default configuration in Huggingface [1]. The F1-scores of "1:2(float)" and "2:4(bfloat16)" without finetuning are obtained by directly using the checkpoints from "Transformer(float)". The F1-scores of "Transformer (float)" and "Transformer (bfloat16)" without finetuning are obtained by directly using the checkpoints from "DFSSATTEN 1:2 (float)"

Table 2: F1 score on BERT-large SQuAD v1.1 (Cl=95%)

| Model | w/o finetune | w/ finetune |
|---|---|---|
| Transformer (float) | $93.22 \pm 0.15$ | $93.17 \pm 0.27$ |
| Transformer (bfloat16) | $93.34 \pm 0.31$ | $93.18 \pm 0.27$ |
| DFSSATTEN 1:2 (float) | $92.86 \pm 0.22$ | $93.07 \pm 0.17$ |
| DFSSATTEN 2:4 (bfloat16) | $93.00 \pm 0.16$ | $\mathbf{93.28 \pm 0.29}$ |

and "DFSSATTEN 2:4 (bfloat16)", respectively, and running inference with dense attention mechanism.

As shown in Table 2, with finetuning, our 1:2 sparsity has only 0.1 F1 score loss that is smaller than the standard deviation. Our 2:4 sparsity even achieves a little bit of performance improvement over the dense baseline. One plausible explanation is that while the 2:4 sparsity can keep most of the important edges, it also occasionally drops a small fraction of important edges which acts like the attention dropout technique (Zehui et al., 2019). Besides, directly applying our methods to the dense transformer without finetuning also achieves comparable results, it justifies that our method can well approximate the dense attention mechanism.

**Masked Language Modeling**. We also evaluate our models on the masked modeling tasks on Wikitext-2 and Wikitext-103 under sequence length 512. Similar to the question-answering tasks, we choose the "roberta-large" as the pretrained model and finetune it under the default configuration in Huggingface [2]. The results are summarized in Table 3. Similarly, the perplexities achieved by our methods are on par with the dense transformer.

Table 3: Perplexity on roBERTa-large (Cl=95%)

| Model | Wikitext-2 | | Wikitext-103 | |
|---|---|---|---|---|
| | w/o finetune | w/ finetune | w/o finetune | w/ finetune |
| Transformer (float) | $2.85 \pm 0.09$ | $\mathbf{2.83 \pm 0.09}$ | $2.63 \pm 0.03$ | $2.62 \pm 0.04$ |
| Transformer (bfloat16) | $2.85 \pm 0.05$ | $2.85 \pm 0.07$ | $2.62 \pm 0.08$ | $2.63 \pm 0.05$ |
| DFSSATTEN 1:2 (float) | $2.88 \pm 0.06$ | $2.88 \pm 0.07$ | $2.64 \pm 0.06$ | $2.64 \pm 0.06$ |
| DFSSATTEN 2:4 (bfloat16) | $2.88 \pm 0.07$ | $2.84 \pm 0.04$ | $2.63 \pm 0.03$ | $\mathbf{2.61 \pm 0.04}$ |

**Long Range Arena**. For sequence length longer than 512, we incorporate four tasks from the Long Range Arena (Tay et al., 2021), including ListOps, Text Classification, Document Retrieval, and Image Classification under sequence lengths 2048, 2048, 4096, and 1024, respectively. We omit the Pathfinder (1K) task as we cannot replicate the results, which was also reported in Lu et al. (2021). For a fair comparison with other efficient transformers, the model is trained from scratch under the default configurations. The results are summarized in Table 4. Our method achieves comparable accuracy on all the three benchmarks for long sequence.

Table 4: Accuracy of different transformer models on LRA benchmark. We follow the training instructions from Tay et al. (2021) to reuse the results from this paper.

| Model | ListOps (n=2048) | Text (n=2048) | Retrieval (n=4000) | Image (n=1024) | Avg |
|---|---|---|---|---|---|
| Transformer (float) | 35.91 | 65.05 | 61.72 | 42.15 | 51.21 |
| Transformer (bfloat16) | 35.92 | 65.03 | 61.73 | 42.17 | 51.21 |
| Local Attention | 15.82 | 52.98 | 53.39 | 41.46 | 40.91 |
| Sparse Trans. | 17.07 | 63.58 | 59.59 | **44.24** | 46.12 |
| Longformer | 35.63 | 62.85 | 56.89 | 42.22 | 49.40 |
| Linformer | 35.70 | 53.94 | 52.27 | 38.56 | 45.12 |
| Reformer | **37.27** | 56.10 | 53.40 | 38.07 | 46.21 |
| Sinkhorn Trans. | 33.67 | 61.20 | 53.83 | 41.23 | 47.48 |
| Synthesizer | 36.99 | 61.68 | 54.67 | 41.61 | 48.74 |
| BigBird | 36.05 | 64.02 | 59.29 | 40.83 | 50.05 |
| Linear Trans. | 16.13 | **65.90** | 53.09 | 42.34 | 44.37 |
| Performer | 18.01 | 65.40 | 53.82 | 42.77 | 45.00 |
| **DFSSATTEN 1:2 (float)** | 36.85 | 64.95 | 61.83 | 42.02 | 51.41 |
| **DFSSATTEN 2:4 (bfloat16)** | 37.19 | 64.91 | **62.26** | 42.31 | **51.67** |

## 5.2 SPEEDUP

In this section, we demonstrate the speedup achieved by our method across different sequence lengths. For a fair comparison, we only show the speedup achieved on the attention mechanism

---

[1]https://github.com/huggingface/transformers/tree/master/examples/pytorch/question-answering

[2]https://github.com/huggingface/transformers/tree/master/examples/pytorch/language-modeling

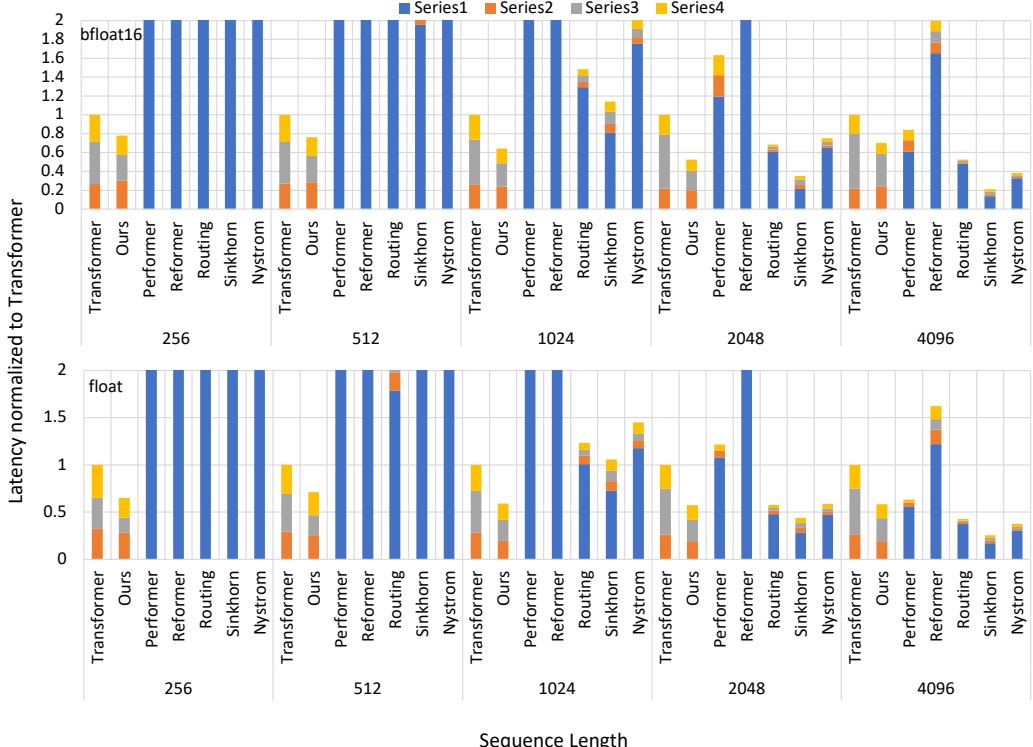

Figure 4: Latency breakdown of different attention mechanism. For each configuration, we normalize the latency to Transformer with full attention mechanism and cut off the axis at 2 for clarity.

declared in equation 1. This is because the end-to-end speedup can be affected by various other factors including the quantization and pruning strategies applied to the other parts of the transformer models, the embedding sizes, and the efficiency of the code implementation (e.g. some operators could be fused for lower latency). We present the end-to-end speedup and memory footprint reduction under different sequence length, number of heads, and hidden dimension in Appendix A.6. For models in previous studies, We also apply the PyTorch JIT script when possible in case that their implementations are not efficient. The configuration we use is as follows: Each layer contains 4 heads, the feature dimension per head is 64. The batch size is set to be large enough to keep the GPU busy. We summarize the profiling results in Figure 4. We normalize the latency to the Transformer with full attention mechanism under each configuration. We also cut the y axis off at 2 for clarity, because some methods designed for long sequence could be more than $20\times$ slower than the dense transformer at moderate sequence length.

First of all, our method achieves $1.27\sim1.89$x speedup over the transformer with full attention. It is the only method that brings consistent speedup across different sequence lengths, while other methods from previous papers suffer from high overhead at moderate and short sequence lengths. Second, under the float data type, our method achieves speedup in all the three stages with zero overhead. This accords with our arguments in Section 3. Under the data type bfloat16, the $\boldsymbol{QK}^T$ in our method is a little bit slower than the dense baseline. The reason is that selecting 2 larger ones from 4 elements requires more comparisons, which results in more warp divergence.

## 6 CONCLUSION AND DISCUSSION

In this paper, we present DFSSATTEN, a dynamic fine-grained structured sparse attention mechanism that dynamically prunes the $\boldsymbol{QK}^T$ on the fly to 1:2 and 2:4 structured sparsity. As it only requires 50% sparsity, it can achieve no accuracy loss compared with the full attention mechanism across tasks in various domains and sequence lengths. Besides, it only requires modifying a few lines of code, which makes it a drop-in replacement of the full-attention mechanism. Moreover, powered by our customized CUDA kernels and the new sparse tensor core on Ampere GPU, we achieve $1.27\sim1.89\times$ speedup over the full attention in arbitrary sequence length. All of these pieces of evidence demonstrate that our method can be a good replacement for the full attention mechanism. Our method is also orthogonal to many existing efficient attention mechanisms and can potentially be applied jointly for further speedup. We present two examples in Appendix A.7.

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

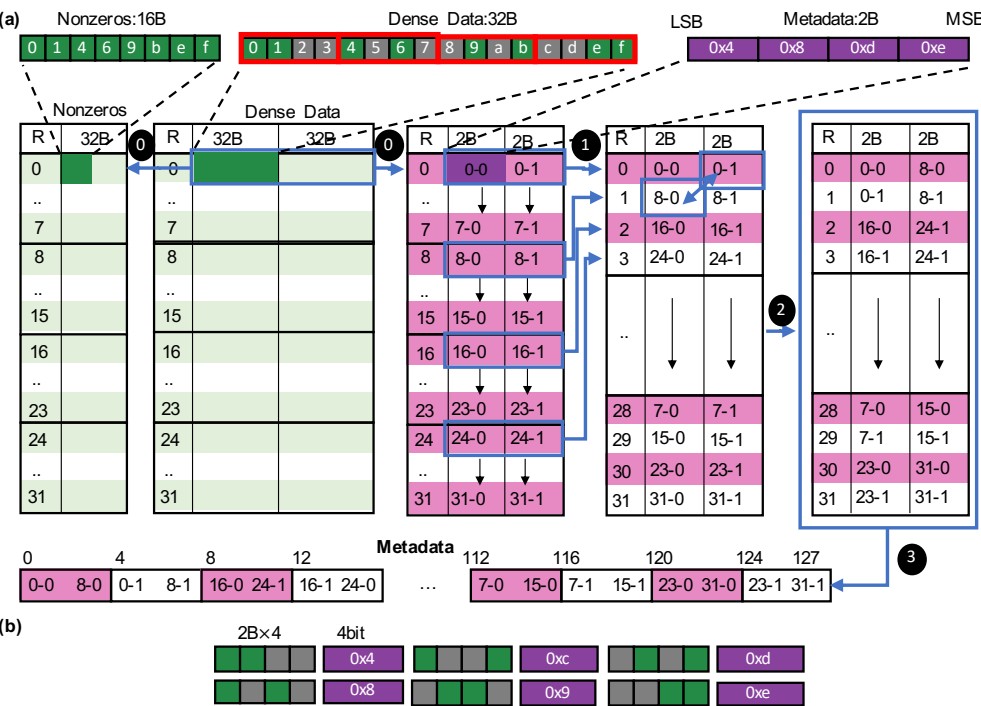

Figure 5: Prune dense data and generate nonzeros, metadata.

# A APPENDIX

## A.1 KERNEL DESIGN DETAILS

In this section, we first demonstrate how a dense matrix is pruned and compressed under the 50% structured sparsity on Ampere GPU in Section A.1.1. Then, we detail the design and implementation of the SDDMM, Softmax, and SpMM kernels in Section A.1.2 and A.1.3.

### A.1.1 STRUCTURED PRUNING OF DENSE MATRIX

We first illustrate how to dynamically prune a dense matrix with 50% structured sparsity. Under data type float, we select the larger one in every two consecutive elements. If the data type is bfloat16, we select two larger ones in every four consecutive elements. We compress the pruned dense matrix to nonzeros and metadata following CUTLASS et al (2021) as there are two benefits. First, it can be directly used by high-performance SpMM kernels in CUTLASS. Second, as we will show in Section A.1.2, it can be dynamically generated from the SDDMM kernel with neglectable overhead.

As shown in Figure 5, the basic tile size to prune is $32 \times 64$-byte, this corresponds to a $32 \times 32$ block under bfloat16 or $32 \times 16$ block under float. There are four major steps: ❶ Prune 50% of each consecutive 8B data, generate nonzeros and metadata; ❷ Interleave the metadata rows by 8; ❸ Switch the metadata along sub-diagonal. ❹ Write metadata and nonzeros to global memory.

In detail, 2 out of 4 2-byte data are select based on their magnitude and a unique 4-bit metadata is assigned to each combination in ❶. The correspondence between selection pattern and metadata is enumerated in Figure 5 (b). Notably, with float32 data type, each 32-bit data occupies two consecutive 2-byte slots. Therefore, it only supports the patterns under *0x4* and *0xe*. After generating the 4-bit metadata, consecutive four of them are concatenated to a 2B metadata block. Then, the rows of metadata are interleaved by 8 in ❶ following

$$dst\_row = \lfloor row/32 \rfloor \times 32 + (row\%8) \times 4 + \lfloor (row\%32)/8 \rfloor. \qquad (9)$$

In ❷, the metadata blocks at upper right and lower left of each $2 \times 2$ grid are switched. At last, in ❸, the metadata produced by ❷ is written into global memory following the interleaved column-

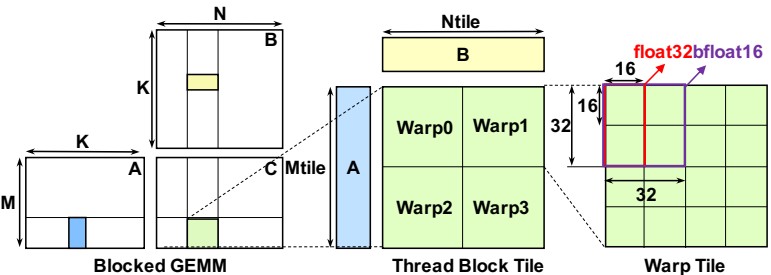

Figure 6: Dense Matrix Multiplication (GEMM) Tiling Design.

major format under stride 4-byte. This can be realized by interpreting two consecutive metadata as an int object and then write it to DRAM in column-major. The nonzeros are simply writen to global memory under row-major.

### A.1.2 SDDMM KERNEL DESIGN

Our strategy for dynamically pruning the attention score matrix has two steps. First, perform a conventional dense GEMM. Second, prune the GEMM output with procedures described in Section A.1.1. However, if the second step is implemented as a separate GPU kernel, we need to write the dense attention score matrix to DRAM and read it back. This not only introduces high overhead, but also prevents us from reducing global memory footprint. To address this issue, we implement the pruning step as an epilogue attached to the conventional GEMM kernel: the results of the dense GEMM are stored in the registers, the epilogue processes the results and then writes nonzeros and metadata to global memory.

**Dense GEMM**. The GEMM step is no different from conventional GEMM kernels, and all the existing optimizations can be used. The tiling is shown in Figure 6: each thread block processes a $Mtile \times Ntiles$ output tile, which is further partitioned to several warp tiles. Each warp tile is composed of a grid of $16 \times 16$ blocks that matches the tensor core output size. In each thread block, all the threads jointly load $Mtile \times Ktile$ and $Ktile \times Ntile$ input tiles from matrix A and B into the shared memory. We use the new synchronize copy feature on Ampere architecture to fully utilize the memory bandwidth and reduce register usage. To fully annihilate shared memory bank conflict, we use the XOR layout. Once the load is completed, the warps fetch their source operands from shared memory with *ldmatrix* and perform a $(16 \times 32B) \cdot (32B \times 16)$ warp matrix multiply accumulate (*wmma*) with tensor core. Notably, float data will be converted to *tensorfloat-32* before *wmma*. To reduce accumulation error, we accumulate the partial sum as float regardless of the source operand data type. Besides, software 2-stage pipeline is used to overlap memory access and computation with double buffering (et al, 2021). Although deeper software pipeline can be built on Ampere, we find 2 stages is enough as the inner-product dimension $K$ is usually very small (e.g. 64). More detailed explanation of the above techniques can be found in this GTC 2020 talk (Kerr, 2020).

**Pruning the GEMM result**. In the pruning step, the warp tile is partitioned to a grid of $32 \times 64B$ blocks that are processed by the warp one at a time.

Under data type float, the register layout of the $32 \times 16$ block is illustrated in Figure 7 (a). It consists of two $16 \times 16$ *wmma* blocks, so each thread has sixteen 32-bit registers to hold the results. The registers are annotated with "T*thread_id*{register_id}". As the adjacent two data are held by the same thread, we can simply compare them and the larger one is retained.

Under data type bfloat16, we need to select 2 larger ones from adjacent 4 entries. However, under the naive mapping shown in Figure 8 (a), these 4 entries are held by 2 thread. Therefore, we need additional warp shuffle to first pass these 4 entries to the same thread, then compare them and obtain the 2 larger ones. This will introduce additional overhead. To solve this problem, we propose to interleave the columns when loading matrix B to shared memory by simply manipulating the pointer to the global memory at the beginning. The resulted mapping to the registers is shown in Figure 8 (b) which is equivalent with Figure 7 (a) bfloat16. After the interleaving, consecutive four

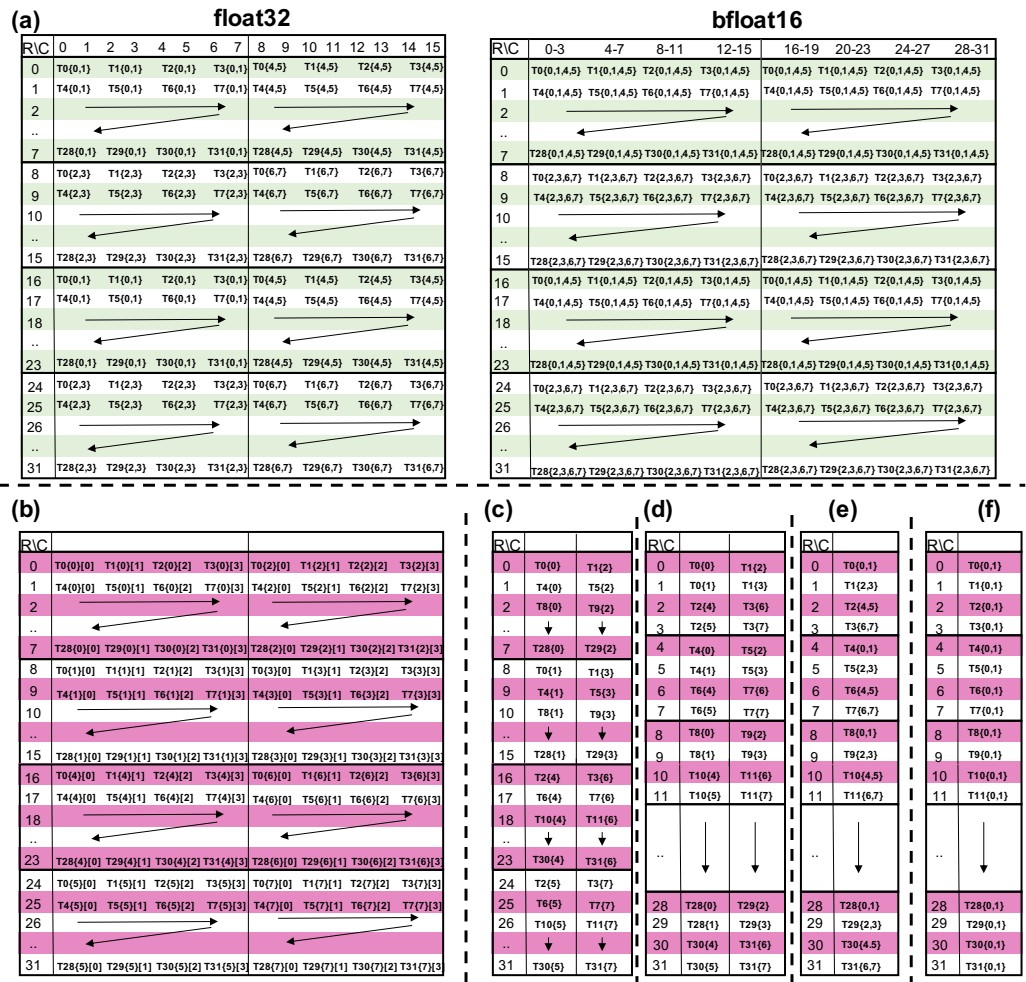

Figure 7: Mapping between the registers and data, metadata.

data are naturally held by the same thread, and we select 2 larger ones from them. To reduce branch divergence, the selection is done by comparing the sum of any two data.

**Generate Metadata and Nonzeros.** For both float and bfloat16 data type, each comparison produces a 4-bit metadata. Next, following the procedures described in Section A.1.1, we need to concatenate consecutive four metadata to a 16-bit metadata block. This is done in two steps. First, put the 4-bit metadata to the correct position of a int16 register with bit shift. Second, share these int16 registers cross threads with warp shuffle, and concatenate them with bitwise OR. As consecutive four metadata are held by thread *4t* to *4t+3*, we put the 4-bit metadata of thread *4t+k* to [$k \times 4$:$k \times 4 + 3$] bits in the *int16* object in the first step. The detailed layout is shown in Figure 8 (b), where we denote each 4-bit metadata as "T*thread_id*{*register_id*}[*bit_id*]". The result of the second step is shown in Figure 7 (c). Figure 7 (d) and (e) illustrate the result after ❶ and ❷ in Figure 5. Notably, these two step only change the logic mapping of the metadata and the register allocation is not affected. So no code is required for these two steps. At last, we need to write the metadata and nonzeros to global memory following ❸ in Figure 5. As shown in Figure 7 (e), each row is held by consecutive two *int16* registers of the same thread, so we can simply reinterpret it as an =*int32* object and write the metadata to global memory in column major. For the nonzeros, we simply coalesce them in the shared memory and then write to global memory in row-major.

**Batched Kernel.** The self-attention layer in transformer usually has multiple independent attention heads. Instead of launching one CUDA kernel for each attention head, using a batched kernel that

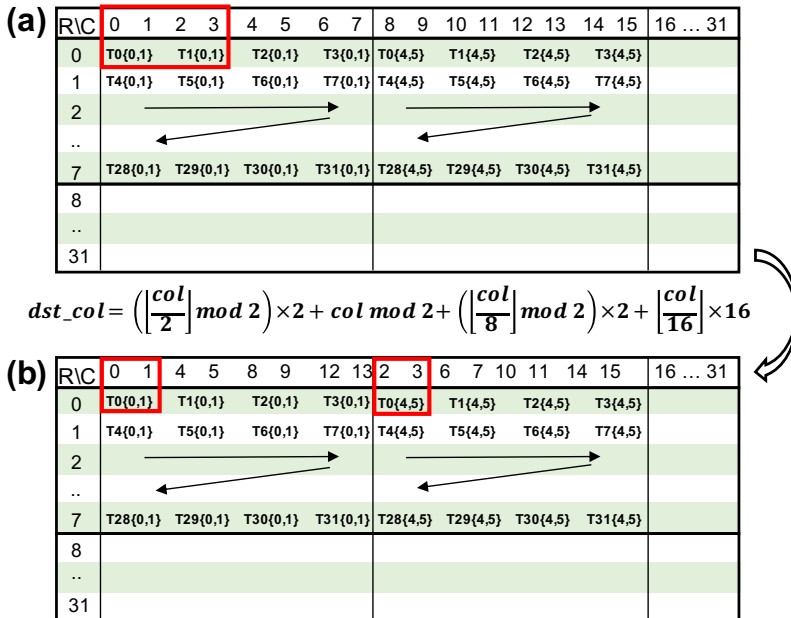

Figure 8: Interleave the columns for matrix B to reduce cross-lane data sharing during pruning for bfloat16.

processes all the heads can better utilize the GPU resources and reduce kernel launching overhead. We support the batched computation by using the *blockIdx.z* to index the heads in the batch and update the pointers to the input and output based on the index.

**Blocked-ELL Sparsity**. Under long sequence length, higher sparsity is desired to reduce computation cost and memory footprint. Our kernel support hybrid blocked-ELL sparsity Zaheer et al. (2020) and 50% structured sparsity. To support this feature, we set the block size in blocked-ELL to the thread block tile size of the GEMM. Therefore, we can simply skip those pruned blocks during the execution.

### A.1.3 Softmax and SpMM Kernel

In this section, we detail the implementation of the softmax and SpMM kernels.

**Softmax Kernel**. To improve numerical stability, the softmax on GPU is computed with

$$softmax(\boldsymbol{x})_i = \frac{e^{x_i - max(\boldsymbol{x})}}{\sum_j e^{x_j - max(\boldsymbol{x})}}. \tag{10}$$

Therefore, each element in $\boldsymbol{x}$ has to be loaded for three times. 1) compute $c = max(\boldsymbol{x})$; 2) compute $s = \sum_j e^{x_j - c}$; 3) compute $e^{x_i - c}/s$. Instead of loading $x_i$ from global memory in each time, we cache it in the register when the whole row fits in the register file capacity. Besides, the ordinary softmax kernel in libraries like PyTorch can also be used.

**SpMM Kernel**. As we encode the nonzeros and metadata following the CUTLASS et al (2021), we directly construct the SpMM kernels from the CUTLASS APIs. To support the hybrid blocked-ELL and structured 50% sparsity, we modify the *PredictedTileAccessIterator* class in CUTLASS to skip the tiles masked out by the blocked-ELL sparsity.

### A.2 Proof of Proposition 4.1

*Proof.* Under the assumption that the entries in $\boldsymbol{Q}\boldsymbol{K}^T/\sqrt{d}$ follow i.i.d. $\mathcal{N}(\mu, \sigma)$, we denote $x_{i,j} = e^{\mu + \sigma z_{i,j}}$, where $z \sim i.i.d.$ $\mathcal{N}(0, 1)$. Then we can substitute it into the definition of the softmax and

get

$$\boldsymbol{A}_{u,v} = \frac{x_{u,v}}{\sum_{i=1}^{n} x_{u,i}}. \tag{11}$$

We substitute the above equation into the definition of $L^p$-Quality and get

$$\mathcal{Q}^p = \frac{1}{n} \sum_{j=1}^{n} \frac{\sum_{i=1}^{n} (\boldsymbol{m} \odot \boldsymbol{A})_{j,i}^p}{\sum_{i=1}^{n} \boldsymbol{A}_{j,i}^p} = \frac{1}{n} \sum_{j=1}^{n} \frac{\frac{1}{n} \sum_{i=1}^{n} m_{j,i} x_{j,i}^p}{\frac{1}{n} \sum_{i=1}^{n} x_{j,i}^p} \tag{12}$$

With $n \to \infty$, the denominator can be approximated with

$$\frac{1}{n} \sum_{i=1}^{n} x_{j,i}^p \approx \int_{-\infty}^{\infty} \frac{e^{p\mu+p\sigma z}}{\sqrt{2\pi}} exp\left(-\frac{z^2}{2}\right) dz = exp\left(p\mu + \frac{p^2\sigma^2}{2}\right) \tag{13}$$

**Top-K Sparsity**. When the sequence is long enough such that we have $n \to \infty$, the numerator can be approximated with

$$\frac{1}{n} \sum_{i=1}^{n} m_{j,i} x_{j,i}^p \approx \int_{\sqrt{2}erfinv(1-2s)}^{\infty} \frac{e^{p\mu+p\sigma z}}{\sqrt{2\pi}} exp\left(-\frac{z^2}{2}\right) dz$$

$$= exp\left(p\mu + \frac{p^2\sigma^2}{2}\right) \frac{1 + erf\left(\frac{p\sigma}{\sqrt{2}} - erfinv(1-2s)\right)}{2} \tag{14}$$

Therefore, the $L^p$-Quality of Top-K sparsity is

$$\mathcal{Q}_{topk}^p \approx \frac{1 + erf\left(\frac{p\sigma}{\sqrt{2}} - erfinv(1-2s)\right)}{2}. \tag{15}$$

**Fixed Sparsity**. Without any assumption on the distribution of important edges in $\boldsymbol{A}$, applying a fixed pattern is equivalent with uniformly sampling with probability $s$ and we have

$$\frac{1}{n} \sum_{i=1}^{n} m_{j,i} x_{j,i}^p \approx exp\left(p\mu + \frac{p^2\sigma^2}{2}\right) s. \tag{16}$$

Therefore, the $L^p$-Quality of the fixed sparsity is

$$\mathcal{Q}_{fix}^p \approx s. \tag{17}$$

**2-to-1 Sparsity**: This sparsity pattern select the larger one in every two elements. We denote adjacent two elements with

$$X = e^{\mu+\sigma Z_1}, Y = e^{\mu+\sigma Z_2}; Z_1, Z_2 \sim \mathcal{N}(0,1), \tag{18}$$

$Z_1$ and $Z_2$ are independent. Then we have

$$\frac{1}{n} \sum_{i=1}^{n} m_{j,i} x_{j,i}^p \approx \frac{1}{2} \mathbb{E}\left[max(X^p, Y^p)\right] =$$

$$\frac{1}{2}\left[\iint_{z_1 \geq z_2} e^{p\mu+p\sigma z_1} \frac{1}{2\pi} exp\left(-\frac{z_1^2+z_2^2}{2}\right) dz_1 dz_2 + \iint_{z_1 < z_2} e^{p\mu+p\sigma z_2} \frac{1}{2\pi} exp\left(-\frac{z_1^2+z_2^2}{2}\right) dz_1 dz_2\right] \tag{19}$$

We denote

$$x = \frac{z_1 - z_2}{\sqrt{2}}, \quad y = \frac{z_1 + z_2}{\sqrt{2}}, \tag{20}$$

then we have

$$\iint_{z_1 \geq z_2} e^{p\mu+p\sigma z_1} \frac{1}{2\pi} exp\left(-\frac{z_1^2+z_2^2}{2}\right) dz_1 dz_2$$

$$= \int_{-\infty}^{\infty} \int_{0}^{\infty} e^{p\mu+\frac{p\sigma}{\sqrt{2}}(x+y)} \frac{1}{2\pi} exp\left(-\frac{x^2+y^2}{2}\right) dx dy$$

$$= \int_{-\infty}^{\infty} \int_{0}^{\infty} e^{p\mu+\frac{p^2\sigma^2}{2}} \frac{1}{2\pi} exp\left[-\frac{1}{2}\left(x - \frac{p\sigma}{\sqrt{2}}\right)^2 - \frac{1}{2}\left(y - \frac{p\sigma}{\sqrt{2}}\right)^2\right] dx dy \tag{21}$$

$$= e^{p\mu+\frac{p^2\sigma^2}{2}} \int_{0}^{\infty} \frac{1}{\sqrt{2\pi}} exp\left[-\frac{1}{2}\left(x - \frac{\sigma}{\sqrt{2}}\right)^2\right] dx = \frac{e^{p\mu+\frac{p^2\sigma^2}{2}}}{2}\left[1 + erf\left(\frac{p\sigma}{2}\right)\right].$$

With the conclusion above, we have

$$\frac{1}{n}\sum_{i=1}^{n} m_{j,i} x_{j,i}^p = exp\left(p\mu + \frac{p^2\sigma^2}{2}\right)\frac{1 + erf\left(\frac{p\sigma}{2}\right)}{2}. \tag{22}$$

The $L^P$-Quality of 1:2 sparsity can be computed with

$$\mathcal{Q}_{1:2}^p = \frac{1 + erf\left(\frac{p\sigma}{2}\right)}{2}. \tag{23}$$

**2:4 Sparsity**: This sparsity pattern select the largest two elements in consecutive four elements. While it is more challenging to find an explicit expression for $\mathcal{Q}_{4-to-2}^p$, a trivial lower bound can be found with

$$\frac{1}{n}\sum_{i=1}^{n} m_{j,i} x_{j,i}^p \approx \frac{1}{4}\mathbb{E}\left[max(X^p + Y^p, X^p + U^p, X^p + Vp, Y^p + U^p, Y^p + V^p, U^p + V^p)\right]$$

$$\geq \frac{1}{4}\left(\mathbb{E}[max(X^p, Y^p)] + \mathbb{E}[max(U^p, V^p)]\right) = \frac{1}{2}\mathbb{E}[max(X^p, Y^p)], \tag{24}$$

where we have

$$X = e^{\mu+\sigma Z_1}, Y = e^{\mu+\sigma Z_2}, U = e^{\mu+\sigma Z_3}, V = e^{\mu+\sigma Z_4}; Z_1, Z_2, Z_3, Z_4 \sim \mathcal{N}(0,1), \tag{25}$$

$Z_1, ..., Z_4$ are independent. Therefore, the lower-bound of $\mathcal{Q}_{2:4}^p$ is

$$\mathcal{Q}_{2:4}^p \geq \frac{1 + erf\left(\frac{p\sigma}{2}\right)}{2}. \tag{26}$$

$\square$

### A.3 PROOF OF PROPOSITION 4.2

*Proof.* First of all, thanks to the Tensor Core in latest GPUs, the latency of matrix multiplication operations, both sparse and dense, are bounded by the memory access. Therefore, instead of counting the number of MACs (multiply-accumulate operations), the amount of memory access is a better metric to estimate the latency.

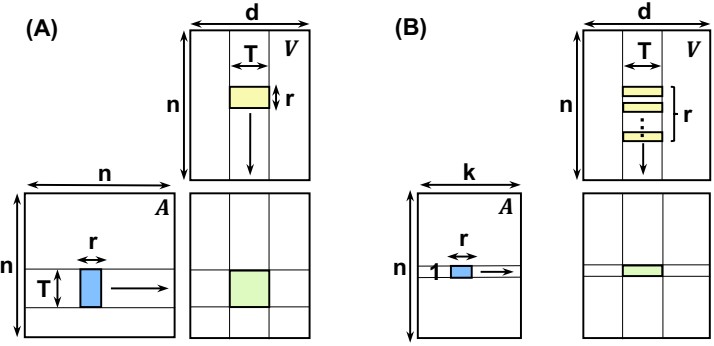

Figure 9: Tiling Matrix-Matrix Multiply

Tiling is a basic optimization applied to optimize matrix matrix multiply on GPU. As shown in Figure 9 (A), the original $n \times n$ output is partitioned to independent blocks with size $T \times T$. When computing each block, operands with size $T \times r$ and $r \times T$ are loaded from $\boldsymbol{A}$ and $\boldsymbol{V}^T$ to the fast memory, respectively. Then, these two operand are multiplied and accumulated to the partial sum stored in the registers. After applying the top-k, as shown in Figure 9 (B), the k elements in each row of $\boldsymbol{A}$ correspond to different rows in $\boldsymbol{A}$. Therefore, we can only partition the output to independent

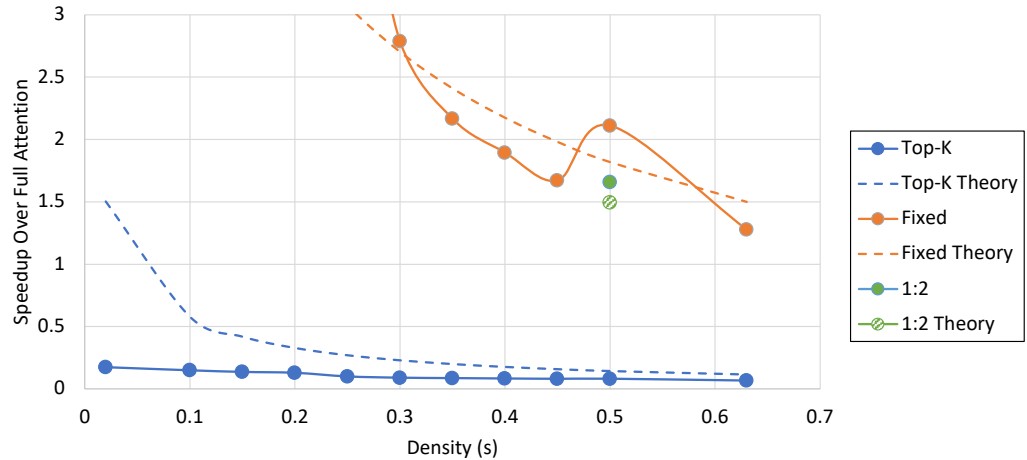

Figure 10: Theoretical and actual speedup achieved by different sparse patterns on A100 GPU.

vectors with size $1 \times T$. During the computation, operands with size $1 \times r$ and $r \times T$ are loaded from $\boldsymbol{A}$ and $\boldsymbol{V}^T$ to the fast memory, respectively. Then, the loaded operands are multiplied and accumulated to the partial sum stored in the registers.

With the tiling strategy mentioned above, we can summarize the amount of memory access in different attentions in the table below.

Table 5: Amount of Memory Access in Different Operations in Attention. $s = k/n$: density of the sparse attention; $T$: tiling size.

|  | $\boldsymbol{QK}^T$ | Softmax | $\boldsymbol{AV}$ |
|---|---|---|---|
| Full Attention | $n^2 \left( \frac{2d}{T} + 1 \right)$ | $2n^2$ | $nd \left( \frac{2n}{T} + 1 \right)$ |
| Explicit Top-k Attention | $n^2 \left( \frac{2d}{T} + 1 \right)$ | $2n^2 s$ | $nd \left( sn + \frac{sn}{T} + 1 \right)$ |

For $\boldsymbol{QK}^T$, as we need to compute all of it before getting the top-k elements, it is a dense matrix matrix multiplication for both full and explicit top-k attention. The Softmax needs to read the $n \times n$ $\boldsymbol{QK}^T$ in, normalizes it, and write the result $\boldsymbol{A}$ back. As the intermediate values can be stored in registers, we only need to count reading $\boldsymbol{QK}^T$ in and writing $\boldsymbol{A}$ out. Therefore, its memory access is $2n^2$ for full attention and $2n^2 s$ for explicit top-k attention. For $\boldsymbol{AV}$ in full attention, the output size is $nd$. As each output element is generated from the inner product between two vectors with length $n$, the total data read equals $nd \times 2n$. However, with the tiling in Figure 9 (A), each operand is reused for $T$ times. Therefore, the total memory access for $\boldsymbol{AV}$ in full attention is $nd(\frac{2n}{T} + 1)$. For $\boldsymbol{AV}$ in explicit top-k attention, as shown in Figure 9 (B), each left-hand-side data is reused for $T$ times while each right-hand-side data is used only for once. Therefore, its memory access equals to $nd(\frac{sn}{T} + sn + 1)$.

The theoretical speedup can be computed with

$$Speedup \leq \frac{n^2 \left( \frac{2d}{T} + 1 \right) + 2n^2 + nd \left( \frac{2n}{T} + 1 \right)}{n^2 \left( \frac{2d}{T} + 1 \right) + 2n^2 s + nd \left( sn + \frac{sn}{T} + 1 \right)} \overset{n \gg d}{\approx} \frac{4d + 3T}{2d + T + (d + 2T + dT)s} \quad (27)$$

$\square$

### A.4 QUALITY OF THE LOTTERY TICKETS UNDER THE SAME EFFICIENCY

In this section, we provide more empirical evidences to support our conclusions. We first compare the theoretical speedup of different sparsity predicted in equation 4, equation 5, and equation 6 and the actual speedup measured on A100 GPU in Figure 10.

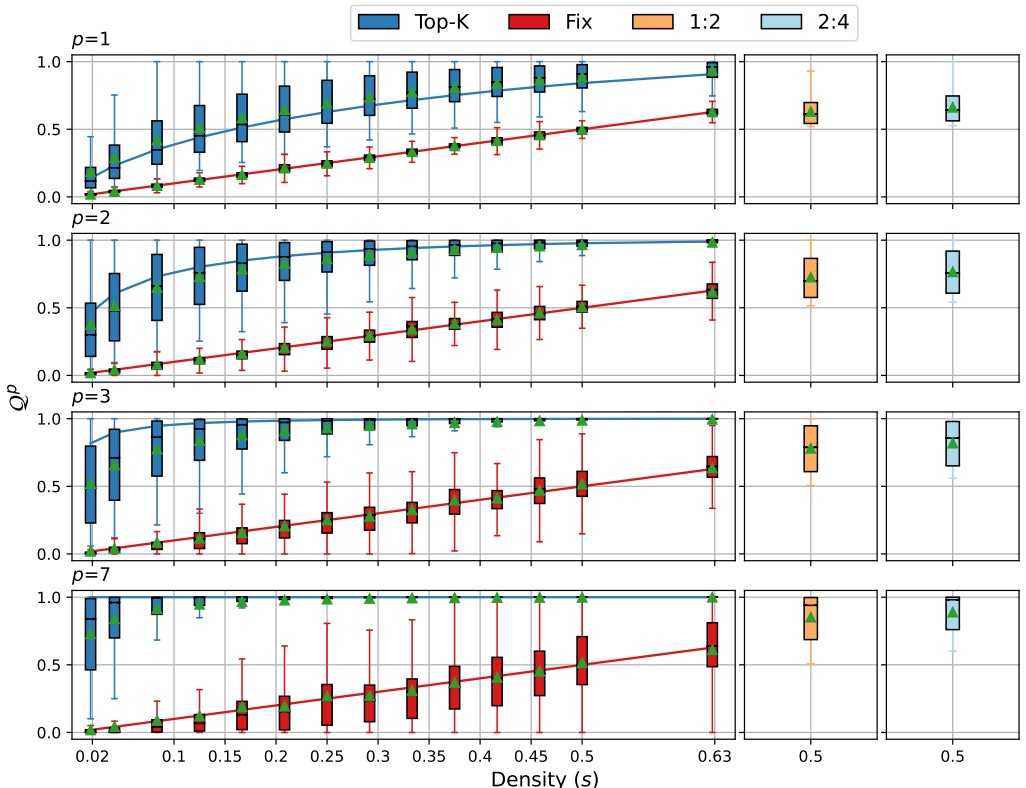

Figure 11: $\mathcal{Q}^p$ under different density $s$ and sparsity strategies. Box plot: Empirical results from BERT-large on SQuAD v1.1; Solid line: Theoretical results from Proposition 4.1.

First of all, the Top-K sparsity is well bounded by the theoretical value, and our method achieves better speedup than the Top-K sparsity when the density $s > 0.02$. This is because gathering top-k elements in each row of the attention weight matrix and sorting them to compressed row format introduce huge overhead.

Second, the speedup achieved by the fixed sparsity is well predicted by our theoretical value. The speedup it achieved is lower than ours when density $s \geq 0.63$, which accords with our theoretical conclusion. Notably, the speedup of fixed sparsity we used here is simply truncate the number of columns of the attention weight matrix based on the density. The actual speedup will be even lower when more fine-grained pattern is involved.

Our method delivers speedup a little bit higher than the theoretical value. This is because the softmax kernel has different implementations under different sequence length. When the sequence length is moderate, as mentioned in Appendix A.3, the data loaded from the attention score matrix can be explicitly cached in fast memory like registers or shared memory for reuse. When sequence length too long for the fast memory to cache, it has to be implicitly reused through lower-level cache or even global memory. The second implementation is slower than the first one as lower-level cache has longer access latency and lower throughput. As our method reduces the sequence length by half, it can use the implementation for moderate sequence length while the full attention is handled by the long sequence version.

In Figure 11, we compute the theoretical value (solid line) and empirical value (box plot) of $\mathcal{Q}^p$ over attention matrix $\boldsymbol{A}$ in BERT-Large on SQuAD v1.1. As $p$ is a task-dependent value that is hard to obtain, we instead sweep through several typical values.

Compared with the top-k sparsity, when $p < 7$, our 1:2 and 2:4 sparsity always achieve better performance than the top-k sparsity when $s < 0.05$. Besides, when $p = 7$, the $\mathcal{Q}^p_{1:2}$ and $\mathcal{Q}^p_{2:4}$ are

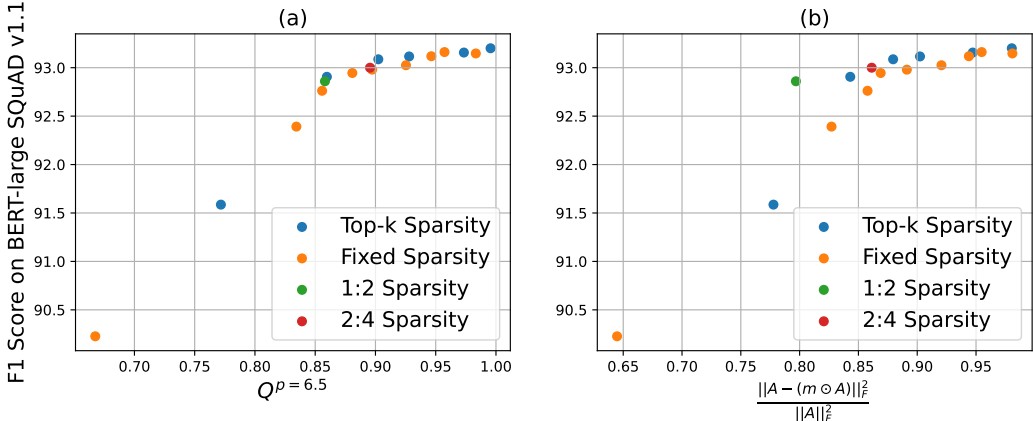

Figure 12: $\mathcal{Q}^p$ under different density $s$ and sparsity strategies. Box plot: Empirical results from BERT-large on SQuAD v1.1; Solid line: Theoretical results from Proposition 4.1.

very close to 1. These observations accord with our conclusion that our 1:2 and 2:4 sparsity can obtain tickets with better quality than Top-K sparsity at the same efficiency.

Compared with the fixed sparsity, our $\mathcal{Q}^p_{1:2}$ and $\mathcal{Q}^p_{2:4}$ are also similar or better than $\mathcal{Q}^p_{fix}$ across different $p$s. This supports our conclusion that our method achieves better performance than the fixed sparsity patterns under the same efficiency.

To show that our $Q^p$ is a good metric to compare the performance of different sparse patterns, we plot the $Q^p$ and F1 score on BERT-large SQuAD v1.1 in Figure 12. As we mentioned before, $p$ is a task-specific value used to model tasks with different degree of dependency on the largest few elements. In order to identify the $p$ for our target task, we tune the value of $p$ until the data points from Top-K sparsity and Fixed sparsity form a monotonically increasing line. We found that $p = 6.5$ is a good choice. This large $p$ accords our observation that the Top-K sparsity works well even under 5.4% density. After anchored the $p$, we put the data points from 1:2 and 2:4 sparsity into the plot and verify if the line is still monotonically increasing. Figure 12 shows that the data points from our 1:2/2:4 sparsity perfectly fills in the monotonically increasing line. Oppositely, The traditional F-norm based metric cannot explain why the 1:2 sparsity has better F1-score than some Fixed Sparsity even though it has lower score. This demonstrates that our $Q^p$ is a better metric than existing metrices.

## A.5 COMPARISON WITH PERFORMER

In this section, we add more discussions on how our method compared with kernel based transformer, i.e. Performer (Choromanski et al., 2021). As our Definition 4.1 is designed to characterize how well the sparse pattern could reserve the important edges in $\boldsymbol{A}$, so it is not suitable for kernel-based attention mechanisms that do not involve sparsity. For example, an approximation of $\boldsymbol{A}$ with high positive approximation error can have $\mathcal{Q}^P \geq 1$ under Definition 4.1. Therefore, we instead compare the mean squared error (MSE) following Choromanski et al. (2021). Given the query and two adjacent key vectors $\boldsymbol{q}$, $\boldsymbol{k}$, and $\boldsymbol{k}' \in \mathcal{N}(0, \boldsymbol{I}_d)$, we denote the softmax kernel between them as $SM(\boldsymbol{q}, \boldsymbol{k}) = exp(\boldsymbol{q}^T \boldsymbol{k}/\sqrt{d})$. And the softmax approximated by our dynamic 1:2 sparsity $\widehat{SM_{1:2}}(\boldsymbol{q}, \boldsymbol{k})$ is defined as

$$\widehat{SM_{1:2}}(\boldsymbol{q}, \boldsymbol{k}) = \begin{cases} exp\left(\frac{\boldsymbol{q}^T \boldsymbol{k}}{\sqrt{d}}\right) & if \ \boldsymbol{q}^T \boldsymbol{k} > \boldsymbol{q}^T \boldsymbol{k}' \\ 0 & else \end{cases}. \tag{28}$$

Then, we can compute its MSE as follows

$$MSE(\widehat{SM_{1:2}}(\boldsymbol{q}, \boldsymbol{k})) = \int_{\boldsymbol{q}^T \boldsymbol{k} < \boldsymbol{q}^T \boldsymbol{k}'} exp\left(\frac{2\boldsymbol{q}^T \boldsymbol{k}}{\sqrt{d}}\right) 2\pi^{-d/2} exp\left(-\frac{||\boldsymbol{k}'||_2^2}{2}\right) d\boldsymbol{k}'. \tag{29}$$

Because $\boldsymbol{q}^T\boldsymbol{k}' = \sum_{i=1}^d q_i k_i'$ is the weighted sum of i.i.d variables following $\mathcal{N}(0,1)$, we have $x = \boldsymbol{q}^T\boldsymbol{k}' \sim \mathcal{N}(0, ||\boldsymbol{q}||_2^2)$. We can substitute it into equation 29 and get

$$
MSE(\widehat{SM_{1:2}}(\boldsymbol{q},\boldsymbol{k})) = exp\left(\frac{2\boldsymbol{q}^T\boldsymbol{k}}{\sqrt{d}}\right) \int_{x>\boldsymbol{q}^T\boldsymbol{k}} \frac{1}{\sqrt{2\pi||\boldsymbol{q}||_2^2}} exp\left(-\frac{x^2}{2||\boldsymbol{q}||_2^2}\right) dx
$$

$$
= exp\left(\frac{2\boldsymbol{q}^T\boldsymbol{k}}{\sqrt{d}}\right) \frac{1 - erf\left(\frac{\boldsymbol{q}^T\boldsymbol{k}}{||\boldsymbol{q}||_2\sqrt{2}}\right)}{2} = SM^2(\boldsymbol{q},\boldsymbol{k}) \frac{1 - erf\left(\frac{\sqrt{d}}{||\boldsymbol{q}||_2\sqrt{2}} ln\left(SM(\boldsymbol{q},\boldsymbol{k})\right)\right)}{2}.
$$

(30)

With Lemma 2 and Theorem 2 in Choromanski et al. (2021), the MSE of their positive softmax kernel with orthogonal random features has an upper bound as follows

$$
MSE\left(\widehat{SM_m^{ort+}}(\boldsymbol{q},\boldsymbol{k})\right) \leq \frac{1}{m} exp\left(\frac{2\boldsymbol{q}^T\boldsymbol{k}}{\sqrt{d}}\right) \left[exp\left(\frac{||\boldsymbol{q}+\boldsymbol{k}||^2}{\sqrt{d}}\right) - 1 - \left(1 - \frac{1}{m}\right)\frac{2}{d+2}\right]
$$

$$
= \frac{1}{m} SM^2(\boldsymbol{q},\boldsymbol{k}) \left[exp\left(\frac{||\boldsymbol{q}||_2^2 + ||\boldsymbol{k}||_2^2}{\sqrt{d}}\right) SM^2(\boldsymbol{q},\boldsymbol{k}) - 1 - \left(1 - \frac{1}{m}\right)\frac{2}{d+2}\right].
$$

(31)

First of all, when $SM(\boldsymbol{q},\boldsymbol{k}) \to 0$, both $MSE(\widehat{SM_{1:2}}(\boldsymbol{q},\boldsymbol{k}))$ and $MSE\left(\widehat{SM_m^{ort+}}(\boldsymbol{q},\boldsymbol{k})\right)$ converge to 0. However, for large $SM(\boldsymbol{q},\boldsymbol{k})$s that are potentially be critical for the model accuracy, the $exp\left(\frac{||\boldsymbol{q}||_2^2 + ||\boldsymbol{k}||_2^2}{\sqrt{d}}\right) SM^2(\boldsymbol{q},\boldsymbol{k})$ term in the positive softmax kernel in Performer could greatly increases the MSE. Oppositely, the $1 - erf\left(\frac{\sqrt{d}}{||\boldsymbol{q}||_2\sqrt{2}} ln\left(SM(\boldsymbol{q},\boldsymbol{k})\right)\right)$ term in our method reduces the MSE. To conclude, while both the positive softmax kernel and ours has low MSE error when approximating small edge weights, our method can better approximate the edges with high magnitude.

From the empirical perspective, as shown in Table 2 and 3, our method can achieve good accuracy even without finetuning. Whereas the Performer still requires tens of thousands steps of finetuning (e.g. Figure 5 in Choromanski et al. (2021)). Table 4 also reveals that Performer has poor accuracy on certain tasks like byte-level document retrieval, while ours consistently achieve accuracy on par with the dense transformer. All this observations suggest that our method can better approximate the full attention mechanism than Performer.

In terms of wall-clock time speedup, Figure 4 illustrates that the Performer can only achieve good speedup at long sequence length. The similar phenomenon is also observed in multiple online forums [3]. Certainly, the PyTorch JIT script does not yield the optimal implementation of the computation graph, but it reveals that tremendous engineering efforts are required for Performer to achieve good speedup under moderate sequence length.

Following Section 4.3, we also compare the theoretical speedup achieved by ours and the Performer.

$$
\boldsymbol{T}_{n\times m}^{(1)} = \frac{\boldsymbol{Q}_{n\times d}}{\sqrt[4]{d}} \boldsymbol{P}_{d\times m}, \quad \boldsymbol{T}_{n\times 1}^{(2)} = \frac{1}{2\sqrt{d}} \sum_{i=1}^d [\boldsymbol{Q}_{n\times d} \odot \boldsymbol{Q}_{n\times d}]_{:,i}
$$

$$
\boldsymbol{T}_{n\times 1}^{(3)} = max_i[\boldsymbol{T}_{n\times m}^{(1)}]_{:i}, \quad \phi(\boldsymbol{Q}_{n\times m}) = \frac{1}{\sqrt{m}} exp\left(\boldsymbol{T}_{n\times m}^{(1)} - \boldsymbol{T}_{n\times 1}^{(2)} - \boldsymbol{T}_{n\times 1}^{(3)} + \epsilon\right)
$$

$$
\boldsymbol{T}_{n\times m}^{(4)} = \frac{\boldsymbol{K}_{n\times d}}{\sqrt[4]{d}} \boldsymbol{P}_{d\times m}, \quad \boldsymbol{T}_{n\times 1}^{(5)} = \frac{1}{2\sqrt{d}} \sum_{i=1}^d [\boldsymbol{K}_{n\times d} \odot \boldsymbol{K}_{n\times d}]_{:,i}
$$

(32)

$$
\boldsymbol{T}_{n\times 1}^{(6)} = max_i[\boldsymbol{T}_{n\times m}^{(4)}]_{:i}, \quad \phi(\boldsymbol{K}_{n\times m}) = \frac{1}{\sqrt{m}} exp\left(\boldsymbol{T}_{n\times m}^{(4)} - \boldsymbol{T}_{n\times 1}^{(5)} - \boldsymbol{T}_{n\times 1}^{(6)} + \epsilon\right)
$$

$$
\boldsymbol{T}_{m\times 1}^{(7)} = \sum_{i=1}^n [\phi(\boldsymbol{K})_{n\times m}]_{i,:}, \quad \boldsymbol{T}_{n\times 1}^{(8)} = 1/\left(\phi(\boldsymbol{Q})_{n\times m} \times \boldsymbol{T}_{m\times 1}^{(7)}\right)
$$

$$
\boldsymbol{T}_{m\times d}^{(9)} = \phi(\boldsymbol{K})_{n\times m}^T \times \boldsymbol{V}_{n\times d}, \quad \boldsymbol{T}_{n\times d}^{(10)} = \phi(\boldsymbol{Q})_{n\times m} \times \boldsymbol{T}_{m\times d}^{(9)} \odot \boldsymbol{T}_{n\times 1}^{(8)}.
$$

The computation steps of Performer are listed in equation 32 where each equation denotes a sub computation graph that can potentially be fused. Notably, this is more complex than the original

---

[3] https://github.com/huggingface/transformers/issues/7675

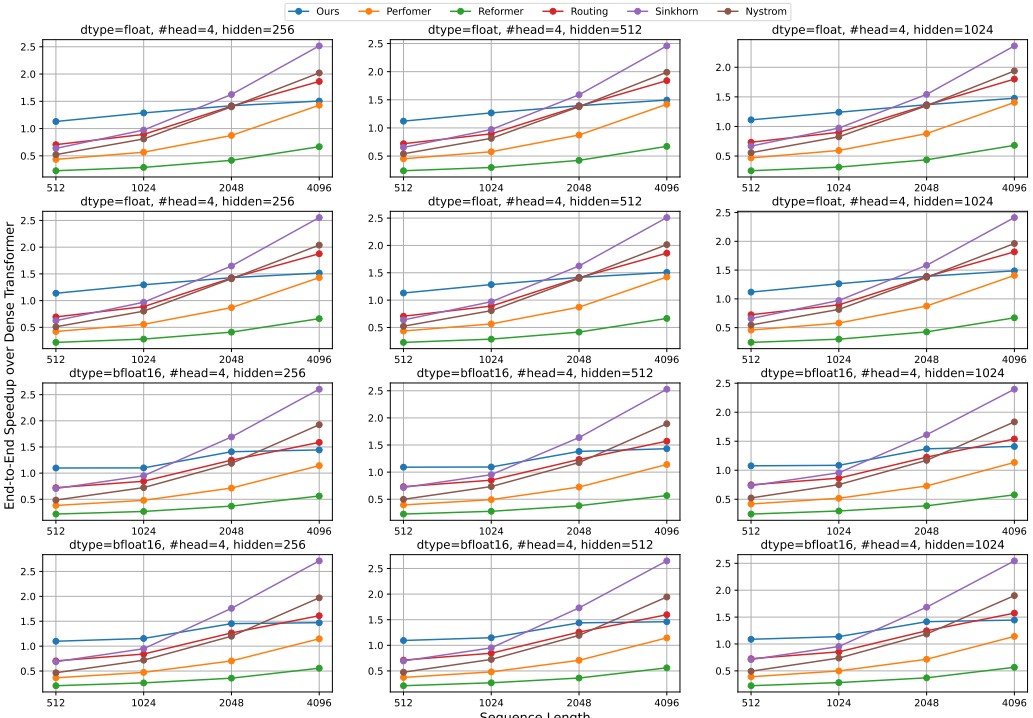

Figure 13: End-to-end inference speedup of different efficient transformers over dense transformer.

mathematical expression to handle the numerical instability of $exp$. The total memory access can be computed with

$$
\begin{aligned}
Speedup = & \left\{ 2 \left[ nm \left( \frac{2d}{T} + 1 \right) + n(d+1) + n(m+1) + n(m+3) \right] + m(n+1) + n\left( \frac{m}{T} + m + 1 \right) \right. \\
& \left. + md \left( \frac{2n}{T} + 1 \right) + nd \left( \frac{2m}{T} + 1 \right) + n \right\} / \left[ n^2 \left( \frac{2d}{T} + 1 \right) + 2n^2 + nd \left( \frac{2n}{T} + 1 \right) \right].
\end{aligned}
\tag{33}
$$

We have $m = dln(d)$ following Theorem 4 in Choromanski et al. (2021). We can substitute $m = 266$, $d = 64$, and $T = 128$ into equation 33 and get $Speedup > 1$ when $n > 672$. On the other hand, the performer achieves the same speedup with ours with $n > 1002$.

To conclude, our method is a good complementary to performer. With delicately optimized computation graph, performer can achieve good speedup and relatively good accuracy under long sequence scenario. In contrary, our method has better speedup and accuracy under moderate and short sequence length. Besides, our method delivers lower approximation error on important edges so it is more friendly to finetuning.

## A.6 END-TO-END SPEEDUP AND MEMORY FOOTPRINT REDUCTION

In this section, we present the end-to-end speedup achieved by our method under different configurations. We use the 4-layer dense transformer model of Text Classification task in Long Range Arena Tay et al. (2021). The dimension of each head is 64. We explore different combination of number of heads (4, 8), sequence length (512, 1024, 2048, 4096), and hidden dimension of the feed forward layer (256, 512, 1024). The end-to-end speedup over the dense transformer under different configurations are plotted in Figure 13.

Our method achieves $1.11 \sim 1.52\times$ and $1.08 \sim 1.47\times$ end-to-end speedup over the dense transformer, it is the only method that deliver end-to-end speedup under all configurations. Under sequence length $\leq 2048$, our method achieves higher speedup than most of the baselines. Although

Sinkhorn transformer (Tay et al., 2020a) has higher speedup than ours at sequence length 2048, as shown in Table 4, its accuracy is less satisfying. This result justifies that our method delivers good speedup under short and moderate sequence length. Notably, this speedup is almost a free lunch. On one hand, Section 5 demonstrates that our method achieves comparable accuracy across different tasks and sequence length, so the model accuracy is not sacrificed. On the other hand, unlike previous efficient transformers, our method has no hyper-parameters and only requires lightweight finetuning process.

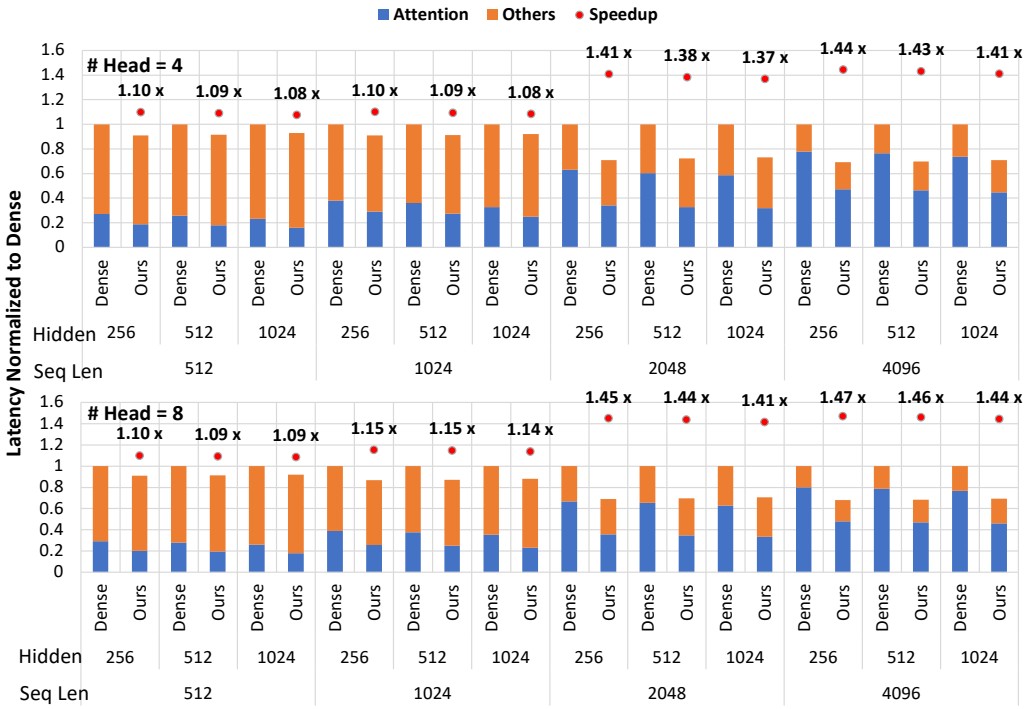

Figure 14: End-to-end inference latency break down under bfloat16.

To study how our method contributes to the end-to-end speedup, we further break down the end-to-end inference time to the attention mechanism and other components under bfloat16. The results are illustrated in Figure 14. Under moderate and short sequence length like 1024 and 512, the "Others" contributes over 70% of the total latency. This is because the size of the matrix multiplications in the feed-forward network and query/key/value projection are comparable with the attention mechanism.

However, unlike the attention mechanism that has limited time budget for compression, the feed-forward network and query/key/value projection use a static weight matrix during inference, so they can be compressed offline. Tons of methods have been proposed in the literature to do that even before the transformers are proposed. For instance, Mishra et al. (2021); Zhou et al. (2020) show that pruning the weights to 2:4 sparsity can deliver $1.3 \sim 1.6\times$ speedup and $2\times$ fewer parameters in the feed-forward and projection layers [4] without accuracy loss on BERT-large. Lagunas et al. (2021) apply structured pruning and achieve $2.4\times$ speedup on SQuAD v1.1 with 1% drop of F1. The MobileBERT (Sun et al., 2020), on the other hand, redesign the network architecture that reduce the hidden dimension of feed-forward network in BERT from 4096 to 512. In terms of quantization, previous work (Zafrir et al., 2019) have shown that the linear layers can be quantized to 8 bit integer.

Besides the linear layers, there are also techniques to accelerate other components in transformers. For instance, the MobileBERT (Sun et al., 2020) replaces the layer normalization to a simple element-wise linear transformation. The input embedding table is also compressed with smaller embedding dimension along with an 1D convolution.

---

[4]https://developer.nvidia.com/blog/exploiting-ampere-structured-sparsity-with-cusparselt/

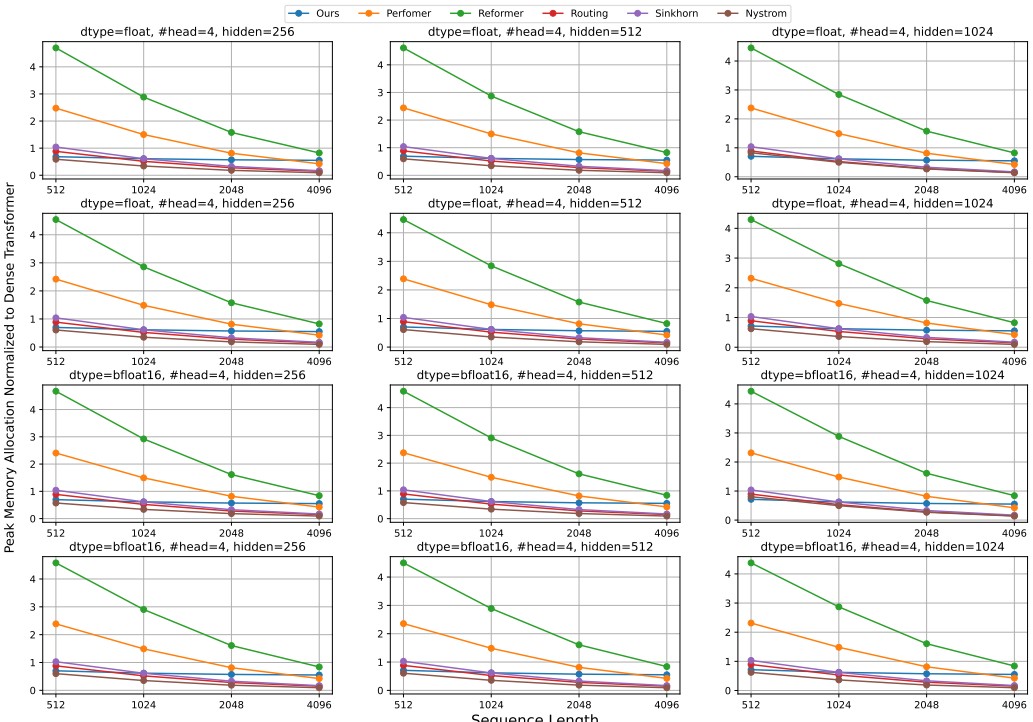

Figure 15: Peak memory allocation normalized to dense transformer under different configurations.

With all these techniques in the literature, it should not be hard to achieve $2\times$ speedup in the non-attention part of transformer models. Then our method could deliver $1.13 \sim 1.41\times$ speedup under sequence length $\leq 1024$.

We also measure the peak memory allocation of different models and configurations, the results are summarized in Figure 15. Our method achieves $1.41 \sim 1.82\times$ memory reduction, which is comparable with or better than most existing efficient transformers when sequence length $\leq 1024$.

## A.7 COMBINATION WITH THE EXISTING EFFICIENT TRANSFORMERS

Existing efficient transformers usually sparsify the full attention mechanism to densely connected clusters (Tay et al., 2020a; Roy et al., 2021; Kitaev et al., 2020; Zaheer et al., 2020) or approximate it with low-rank projection (Wang et al., 2020). As our method is a good approximation of the full attention mechanism and brings wall time speedup at arbtrary sequence length, it can potentially be combined with the existing efficient transformers.

We first demonstrate the combination of our method with Xiong et al. (2021). Xiong et al. (2021) propose a Nystrom-based self-attention mechanism that approximate standard self-attention with $O(n)$ complexity. The Nystromformer is illustrated in Figure 16. We observe that the computation circled in Figure 16 is identical to the standard attention mechanism, so it can be further accelerated with our method. More importantly, the two matrix multipliation involved are the two of the three largest $m \times n$ matrices. It will be very beneficial to reduce their complexity.

We report the accuracy on Image (1K) on LRA (Tay et al., 2021) in Table 6. We first pretrain a standard Nystromformer from the scratch for 35,000 iterations following Xiong et al. (2021). Then, we finetune it for 3,500 iterations (1/10 of the training process) under standard Nystromformer, Nystromformer + DFSSATTEN 1:2, and Nystromformer + DFSSATTEN 2:4. It is obvious that by combining DFSSATTEN and Nystromformer, we can achieve higher accuracy on LRA with lightweight finetuning.

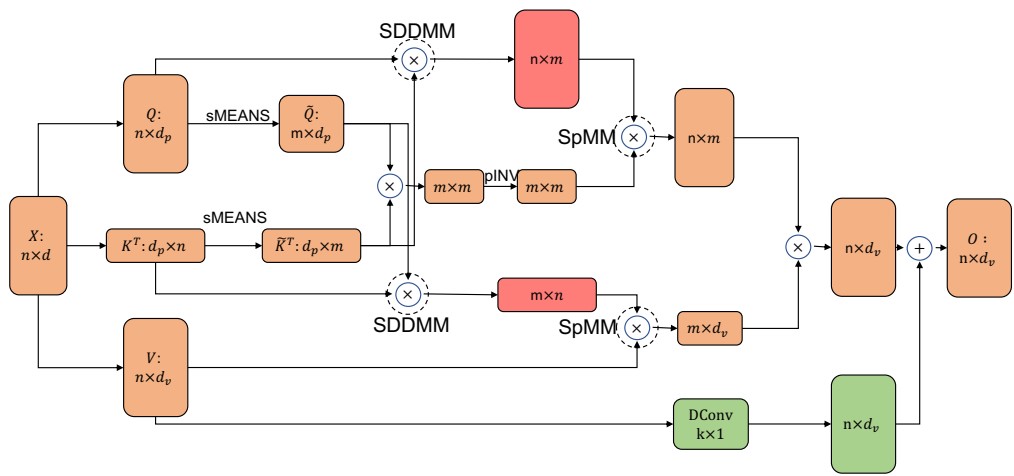

Figure 16: Combination of our method with Nystromformer (Xiong et al., 2021). The two red matrices are stored under 1:2/2:4 structured sparsity.

Table 6: Accuracy on Image (1K) on LRA (Tay et al., 2021) under the combination of DFSSATTEN and Nystromformer (Xiong et al., 2021).

|  | Pretraining | Finetuning |
|---|---|---|
| Nystromformer (float) | 41.17 | 41.52 |
| Nystromformer (bfloat16) | - | 41.59 |
| Nystromformer + DFSSATTEN 1:2 (float) | - | 41.91 |
| Nystromformer + DFSSATTEN 2:4 (bfloat16) | - | **42.54** |

Then we provide a complexity analysis of the combination following Xiong et al. (2021). The landmark selection with segment-means takes $O(n)$, iterative approximation of the pseudoinverse takes $O(m^3)$. The matrix multiplication complexity of the standard Nystromformer takes $O(nm^2 + mnd_v + m^3 + nmd_v)$. After applying our method, it can be reduced to $O(\frac{nm^2}{2} + \frac{nmd_v}{2} + m^3 + nmd_v)$. The memory footprint can be reduced from $O(md_q + nm + m^2 + nm + nd_v)$ to $O(md_q + nm + m^2 + nd_v)$. Given $n \gg m > d_v \approx d_p$, this could be a significant improvement that allows us to use more landmarks $m$ to better approximate the full attention mechanism.

Besides Nystromformer, we also illustrate two possible combinations with BigBird (Zaheer et al., 2020) and Linformer Wang et al. (2020) that can be explored in the future work.

As shown in Figure 17 (A), Zaheer et al. (2020) use block sparsity with block size 64 and compute a full attention within each block. We can apply the 1:2 or 2:4 sparsity within each block to bring further speedup.

Figure 17 (B) gives another example on how to combine our method with Linformer (Wang et al., 2020). Linformer uses low-rank approximation on the attention mechanism as follows:

$$\boldsymbol{O} = softmax\left(\frac{\boldsymbol{Q}\left(\boldsymbol{EK}\right)^T}{\sqrt{d}}\right)\boldsymbol{FV},$$ (34)

where $\boldsymbol{E}, \boldsymbol{F} \in \mathbb{R}^{n \times k}$ are linear projection matrices and $k \ll n$. We can first prune $\boldsymbol{E}$ and $\boldsymbol{F}$ along with other weight matrices to have 1:2 or 2:4 sparsity offline following Mishra et al. (2021). Then we compute $\boldsymbol{EK}$ and $\boldsymbol{FV}$ with Sparse Matrix-Matrix multiplication. Next, we multiply $\boldsymbol{Q}$ and $\left(\boldsymbol{EK}\right)^T$ and the result is pruned to 50% structured fine-grained sparsity on the fly. After applying softmax to the nonzeros, we multiply it with $\boldsymbol{FV}$.

## A.8 VISUALIZE ATTENTION DISTRIBUTION

To illustrate that our DFSSATTEN can well capture the fine-grained sparsity in attention, we visualize the attention weight matrices in BERT-large on SQuAD v1.1 in Figure 18. In detail, we

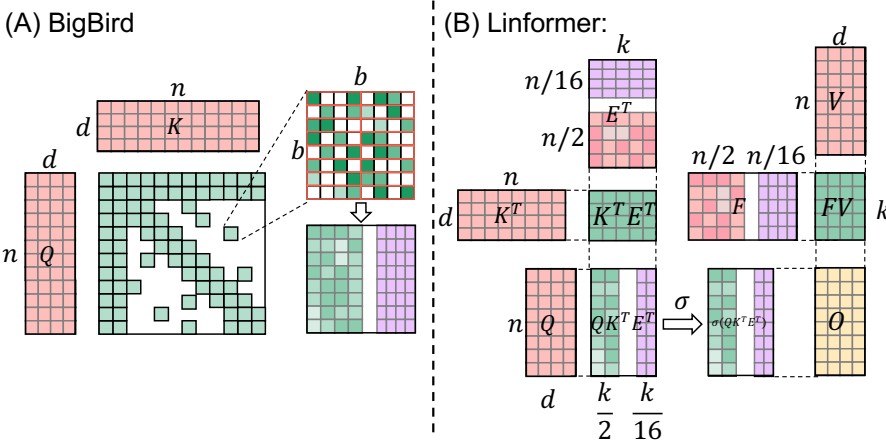

Figure 17: Combination of our method with BigBird (Zaheer et al., 2020) and Linformer (Wang et al., 2020)

run inference of the same input sample in BERT-large model pretrained under dense, 1:2, and 2:4 settings, then collect the attention weight matrix in the first layer. It is obvious that the pattern in dense transformer and our DFSSATTEN are quite similar. The magnitude of nonzero values in DF-SSATTEN are a little bit higher than dense attention. This is because the softmax normalizes the values in each row with the exponential sum of each entry. After removing 50% smaller entries, the magnitude of remaining entries would be relatively higher. Nevertheless, we find that this does not influence the model accuracy, as the forthcoming normalization layers will take care of it.

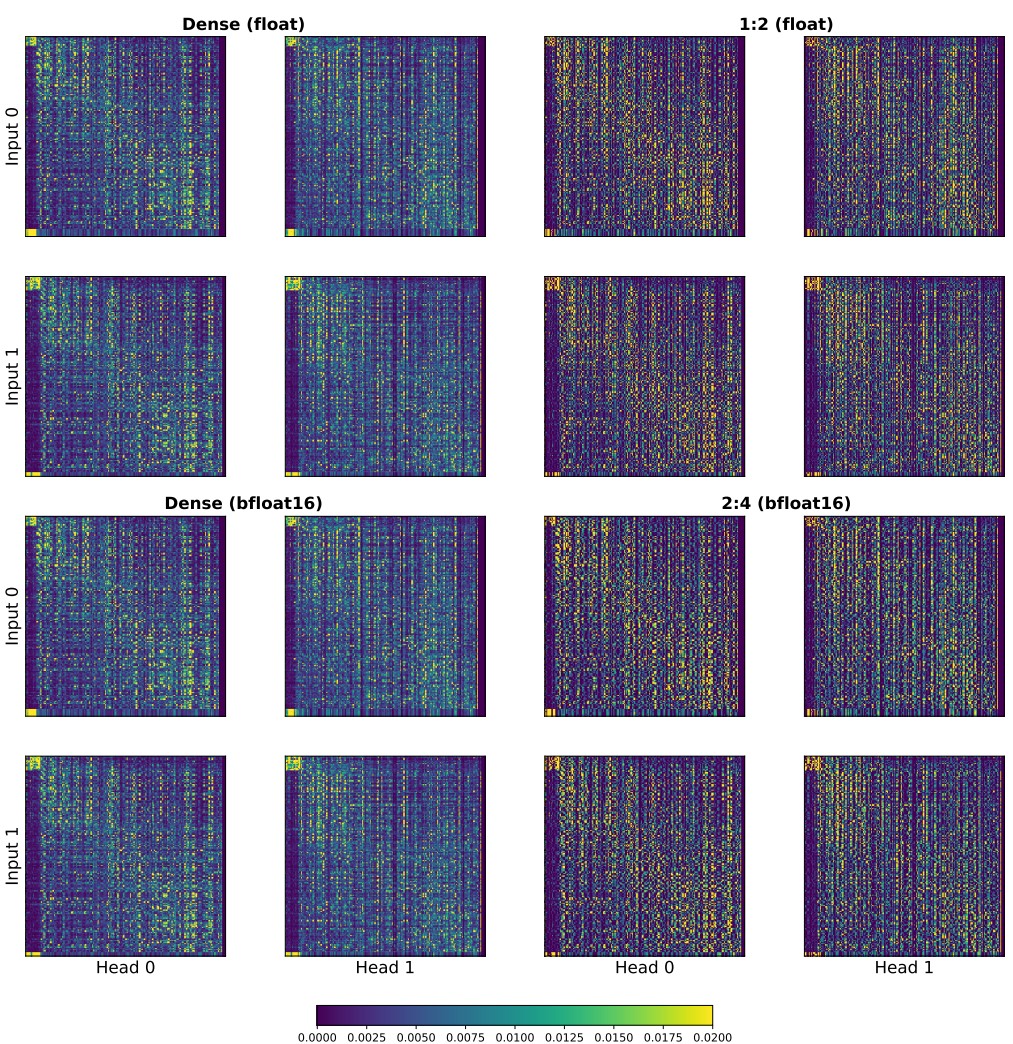

Figure 18: Visualization of attention weight in dense transformer and DFSSATTEN

