# OpenReview forum: "DFSSATTEN:  Dynamic Fine-grained Structured Sparse Attention Mechanism"
_ICLR.cc/2022/Conference — ICLR 2022 Submitted_

### Official Review · Reviewer_kbgy · 2021-10-28

**Correctness:** 4
**Technical Novelty And Significance:** 3
**Empirical Novelty And Significance:** 3
**Recommendation:** 8
**Confidence:** 5

**Main Review:**

pros:

The motivation of this paper, accelerating and maintaining performance using N:M fine-grained structured dynamic sparse attention, is clear and persuasive.

The authors provide the source code with clear comments.

The paper is well-written and easily follow for industrial deployment.

cons:

In section3.1, the reason why this paper started from stage1 confused me, Chen focus on Nvidia Volta Architecture without sparse tensor core, this paper conduct experiments on Ampere Architecture.
what about the speed of QK^T with N:M structured sparse on Ampere Architecture?

How about backward function during finetune sparse models?

Related work, https://arxiv.org/abs/2102.04010 which was the first research paper to train N:M fine-grained static sparse weight from scratch  (including transformer) should be mentioned.

Under bf16 data type, QK^T is slower than the dense baseline? So, according to the reasons, the speedup of dynamic sparse attention may be difficult with a higher sparse ratio (e.g., 2:8) in the future?


**Summary Of The Paper:**

This paper focus on the dynamic N:M fine-grained structured sparse attention implementation on transformers.
Firstly, the authors analyze the theoretical efficiency of Top-K Sparsity, Fixed sparsity and dynamic 1:2/2:4 sparsity,
which demonstrates the dynamic N:M sparse attention can achieve considerable speedup and high-quality approximation.

Then, the authors evaluate the accuracy and the practical speed of the transformer model from the Huggingface model Zoo using the DFSSATTEN module and explain the detailed implementation using the cutlass library.

**Summary Of The Review:**

The paper presents dynamic sparse attention to accelerate the attention module. I give a weak acceptance due to the weakness, I may raise my score according to rebuttal.

---

> ### Author Response · Authors · 2021-11-19
> **Response to Reviewer kbgy (Part 1)**
>
> Thank you for your insightful comments. The detailed responses are summarized as follows.
>
> **(1) In section3.1, the reason why this paper started from stage1 confused me, Chen focus on Nvidia Volta Architecture without sparse tensor core, this paper conduct experiments on Ampere Architecture. what about the speed of QK^T with N:M structured sparse on Ampere Architecture?**
>
> **Response**: This is a good question! There are two major reasons to start from stage 1. First, the SDDMM is hard to achieve speedup over dense matrix multiplication. Second, starting from stage 0 would add unacceptable overhead and hyper-parameters in order to obtain a good prediction of the attention weight matrix.
>
> Particularly, for the first reason, the difficulties in achieving speedup with SDDMM are more severe on Ampere than Volta. We measure the SDDMM on Ampere and find that it is 2x slower than the dense matrix multiplication even with only 2% density. This is because Ampere has a bag of new features like the new generation tensor core, a special data path that loads data from shared memory to registers, and 2x larger shared memory. All these new features target faster dense matrix multiplication and they cannot be leveraged by the SDDMM. In detail, the sparse tensor core you mentioned can only multiply a sparse matrix with a dense matrix, it cannot multiply two dense matrices and produce a sparse one. The new data path “ldmatrix” loads a block of data from the shared memory to register under the data layout required by the tensor core. As fine-grained SDDMM cannot use tensor core, this feature can also not be used. At last, although the shared memory on Ampere is enlarged, the data reuse rate of SDDMM is too low to fully leverage it. Actually, [2] shows that it is better to skip the shared memory in SDDMM for lower memory access latency.
>
> The N:M structured sparsity can also not be used for the SDDMM kernel either. The SDDMM kernel is bounded by memory bandwidth: the computation units are idled most of the time waiting for the operands loaded from memory. With small N and M, we still need to load all rows and columns in the source matrices, and no memory access can be skipped. On the other hand, as there is no execution unit on current GPUs that support multiplying two dense matrices and producing a sparse one, the warp divergence overhead can easily eliminate the benefit from skipping computations. So the optimal way is to compute a dense matrix multiplication, and then only write the preserved data back to global memory. This is exactly what we do.
>
> **(2) How about backward function during finetune sparse models?**
>
> **Response**: Our DFSSATTEN targets accelerating inference. During the finetuning, we emulate the pruning with max-pooling and scatter in the pytorch API. The backward function is automatically handled by the pytorch autograd. This introduces around 5.5% end-to-end overhead to the finetuning process. As our finetuning process is very lightweight (around 30 minutes to finetune BERT-large on SQuAD v1.1), this overhead is acceptable. Oppositely, other efficient transformers take days for retraining or training from scratch.
>
> **(3) Related work, https://arxiv.org/abs/2102.04010 which was the first research paper to train N:M fine-grained static sparse weight from scratch (including transformer) should be mentioned.**
>
> **Response**: Thank you for your recommendation. We have cited this paper in Appendix A.6.
> While both our paper and [1] use the 2:4 fine-grained sparsity, they are quite different. This paper [1] applies the N:M fine-grained sparsity on the static weights. Their major contribution is fusing the pretraining and retraining in NVIDIA Apex into a single stage, rather than achieving real speedup in each iteration during training.
>
> On the other hand, our method is the first one that applies the 2:4 fine-grained sparsity on matrices dynamically generated during inference, which is a more challenging task. This is because static weight can be pruned offline, and the pruning overhead does not occur during inference. Oppositely, the attention weight matrix is dynamically generated for each sample during inference. So any overhead can easily offset the benefit of pruning, and dedicated design for both algorithm and lower-level implementation on GPU are required.

---

> > ### Author Response · Authors · 2021-11-19
> > **Response to Reviewer kbgy (Part 2)**
> >
> > **(4) Under bf16 data type, QK^T is slower than the dense baseline? So, according to the reasons, the speedup of dynamic sparse attention may be difficult with a higher sparse ratio (e.g., 2:8) in the future?**
> >
> > **Response**: First of all, our result shows that 2:4 is enough to achieve not accuracy loss and good speedup. For a higher sparse ratio like 2:8, although more overhead will be added to QK^T, there will also be higher speedup in softmax and AV compensates for it.
> >
> > Besides, the overhead in QK^T is also addressable. Under 2:4 sparsity, there are totally 6 different combinations. To find the combination with the largest sum, 5 comparisons are required for each thread. Our current implementation is based on 5 if-else statements, which introduce many warp divergences that slow down the computation. However, this warp divergence can be easily eliminated with a lightweight hardware feature that already exists in AMD GCN GPUs.
> >
> > AMD GCN ISA exposes an instruction called V_CNDMASK_B32 (http://developer.amd.com/wordpress/media/2013/12/AMD_GCN3_Instruction_Set_Architecture_rev1.1.pdf, page 12-61). This instruction selects between two source registers based on a single condition bit without warp divergence. With an instruction like this, we can eliminate the divergences with a few fixed latency instructions. The latency of these fixed latency instructions can be easily hidden into the memory access timing slots.
> >
> > For a larger volume of combinations, it may also be plausible to add a special execution unit in the SM cores for each lane, which is composed of a set of multiplexers and comparators that select the top-N element in M inputs.
> >
> > **Reference**:
> >
> > [1] Zhou, Aojun, et al. "Learning N: M Fine-grained Structured Sparse Neural Networks From Scratch." International Conference on Learning Representations. 2020.
> >
> > [2] Gale, Trevor, et al. "Sparse GPU kernels for deep learning." SC20: International Conference for High Performance Computing, Networking, Storage and Analysis. IEEE, 2020.

---

> > > ### Comment · Reviewer_kbgy · 2021-11-20
> > > **After Rebuttal**
> > >
> > > Thank you for your detailed and thoughtful response;  the responses addressed all my questions and provided additional context in the revised paper. I raised my score with acceptance.

---

> > > > ### Author Response · Authors · 2021-11-22
> > > > **Thank you!**
> > > >
> > > > We greatly appreciate your acknowledgment!

---

### Official Review · Reviewer_zwV3 · 2021-11-02

**Correctness:** 4
**Technical Novelty And Significance:** 3
**Empirical Novelty And Significance:** 3
**Recommendation:** 8
**Confidence:** 4

**Main Review:**

Strengths:
 - Overall, I found the work to be interesting and the speed-ups quite significant especially for moderately long sequences ($\lt 512$) where most efficient attention mechanisms struggle due to the overheads involved.
 - The paper is well structured and clearly written. The results relating to the quality of approximation / lottery tickets in conjunction with the theoretical speed-ups also shed some light on when different sparsity patterns should be used.
 - The experiments on LRA show that proposed method can train from scratch with no loss in performance while the results on MLM tasks show that the DFSSATTEN can be finetuned from a model pre-trained with full attention. Furthermore, results from Table 1, 2, and 3 also demonstrate that the DFSSATTEN can well approximate pre-trained model with no finetuning.

Some questions and concerns:
 - While the empirical evidence showing that the proposed structured sparse attention can approximate the full attention is very convincing, I am not sure how well does the assumption $\frac{QK^T}{\sqrt{d}}$ follows i.i.d.  $\mathcal{N}(\mu, \sigma)$ (proposition 4.1) hold in practice? This is important as a lot of the claims regarding the quality are hinged on this.
 - The results from Table 2 and 3 (without finetuning) show that DFSSATTEN can be used as drop-in plug in during inference time. It would also be interesting to see what happens if full attention is used on model finetuned with DFSSATTEN, this would help to understand if the DFSSATTEN and full attention learn similar patterns. It would also be useful to compare attention distribution for models finetuned with full and DFSSATTEN.
 - Speedup: From equation (6), for DFSS attention, the speed up would be $\approx$1.5X for long sequences, i.e., when $n \gg d$, however the actual speed ups reported seem to be around 1.87X for long sequences. I am concerned if using only memory access for theoretical estimates is a good approximation.  It would be good to have a plot similar to figure 4 comparing random sparsity, and fix sparsity for few different values of $s$.
 - Figure 10, It would also be nice to add a row showing the actual Bert accuracy plots for different sparsity.
 - How was the fixed sparsity pattern selected in figure 10?
 - Could you please elaborate on the sentence "On the other hand, we have theoretical value of $\sigma \approx 1$ and $p \ge 1$ based on the observation that the edges with higher magnitude are more influential." Specifically why $\sigma \approx 1$ if higher magnitude edges are more influential?

Minor issues:
 - Page 16, before eq. 17, Therefore, $L^p$ quality of ~Top-K~ fixed sparsity.
 - References missing in the table 4

**Summary Of The Paper:**

The paper proposes a method that exploits the 50% structured sparsity supported by tensor cores on modern GPUs. To do so attention scores ($QK^T$) are dynamically pruned such that only 50% attention scores are kept. This improves the computation required for softmax as well as matrix multiplication between pruned attention weight $A$ and value matrix $V$. The authors evaluate the proposed method on LRA and MLM tasks to show that the method can provide practical speed-ups especially for moderately long sequence lengths where other proposed efficient attention mechanisms struggle due to overheads.

**Summary Of The Review:**

The paper showcases speed-ups for a large range of sequences as well as the ability to do both training from scratch and finetuning with no loss in performance. Moreover its compatibility with other efficient attention mechanisms makes it a good choice for a wide range of applications.

---

> ### Author Response · Authors · 2021-11-19
> **Response to Reviewer zwV3 (Part 1)**
>
> Thank you for your positive feedback! We have carefully revised our paper. The detailed responses are summarized as follows.
>
> **(1) While the empirical evidence showing that the proposed structured sparse attention can approximate the full attention is very convincing, I am not sure how well does the assumption $\frac{QK^T}{\sqrt{d}}$ follows i.i.d. $N(\mu, \sigma^2)$ (proposition 4.1) hold in practice? This is important as a lot of the claims regarding the quality are hinged on this.**
>
> **Response**: The i.i.d. normal distribution seems working well in our paper. Indeed, whether the i.i.d. Normal distribution assumption holds is a question shared by almost all theoretical studies in deep neural networks, as there is not a mathematical tool that can precisely model the correlation and distribution of the real-world data in neural networks. Therefore, to justify the correctness of our theory in reality, we conducted numerical studies in Appendix A.5. We plot the theoretical value and empirical value of $Q^p$ in original Figure 10 (now Figure 11). The empirical results are directly obtained from BERT-large on SQuAD v1.1. When p =1 and 2, the theoretical results are very close to the empirical values. They only start to depart when p is large, which is because the few largest values under large p would introduce huge variance to the results, and the sequence length in BERT is not long enough to smooth it.
>
> More importantly, our main theoretical result is that our method has higher $Q^p$ than Top-K and Fix sparsity under the same speedup. This is also supported by our empirical studies. On one hand, Figure 10 of the revised paper shows that our method achieves higher speedup than Top-K and fixed sparsity at 2% and 63% density, respectively. On the other hand, Figure 11 of the revised paper (originally Figure 10) shows that our method has higher $Q^p$ than Top-K and fixed sparsity at 2% and 63% density. Therefore, our theoretical conclusion also holds in reality.
>
> **(2) The results from Table 2 and 3 (without finetuning) show that DFSSATTEN can be used as drop-in plug in during inference time. It would also be interesting to see what happens if full attention is used on model finetuned with DFSSATTEN, this would help to understand if the DFSSATTEN and full attention learn similar patterns. It would also be useful to compare attention distribution for models finetuned with full and DFSSATTEN.**
>
> **Response**: Following your suggestion, we evaluate the full attention on the model finetuned with DFSSATTEN and report the results in Table 2 & 3. The results are very close to the evaluation results on checkpoints finetuned with dense attention, which suggests that DFSSATTEN and full attention learn similar patterns.
>
> We also visualize the attention distribution in Appendix A.8 Figure 17. Our 1:2 and 2:4 sparsity have a very similar distribution with DFSSATTEN.

---

> > ### Author Response · Authors · 2021-11-19
> > **Response to Reviewer zwV3 (Part 2)**
> >
> > **(3) Speedup: From equation (6), for DFSS attention, the speed up would be $\approx 1.5X$ for long sequences, i.e., when $m\gg d$, however the actual speed ups reported seem to be around 1.87X for long sequences. I am concerned if using only memory access for theoretical estimates is a good approximation. It would be good to have a plot similar to figure 4 comparing random sparsity, and fix sparsity for few different values of $s$**
> >
> > **Response**: Following your suggestion, we add a new Figure 10 to the revised paper that compares the theoretical and actual speedup achieved by different sparse patterns under different density s. The actual speedup of Top-K is well bounded by the theoretical value, this is because gathering top-k scores in each row and sorting the indices to construct the compressed row format introduces huge overhead. On the other hand, the theoretical speedup of the fixed sparse pattern well predicts the actual speedup. More importantly, the result justifies two key conclusions we used in our theoretical analysis: our method has higher speedup than top-k sparsity at s=0.02 and fixed sparsity at s=0.63.
> >
> > Previous studies use metrics like FLOPs or asymptotic complexity to model speedup. These metrics are even more inaccurate. On GPUs, the operators in transformers are memory-bound: the computing units are idled most time waiting for operands fetched from memory. Therefore, a kernel with much lower FLOPs may be much slower than a high-FLOPs kernel with fewer memory accesses. This is also the reason why so many fine-grained pruning methods with >90% sparsity are not deployed on GPUs. Our method takes a step forward by taking the actual compute and data reuse pattern on GPU into consideration.
> >
> > Precisely modeling performance on GPU is a challenging research problem. The 20% error is actually pretty good for our lightweight analytic model. [1] propose a performance model that dedicatedly models the performance of different operators in neural networks on GPU. The model considers extensive fine-grained hardware features. Even with this dedicated and complex model, the largest prediction error can still be 100%. One reason behind this is that the implementation has a huge impact on the GPU kernel performance. For instance, simply changing the data layout in the shared memory of a dense matrix multiplication kernel can bring 2x speedup. As deep learning libraries like cuDNN are closed source, it is difficult to precisely model the latency of each operator.
> >
> > **(4) Figure 10, It would also be nice to add a row showing the actual Bert accuracy plots for different sparsity.**
> >
> > **Response**: Thank you for your suggestion. However, it is difficult to directly add BERT accuracy to the original Figure 10 (now Figure 11). This is because the F1 score has a different range from the $Q^p$. And they may not be linearly related.
> >
> > Instead, we find that a better way is to plot the $Q^p$~F1 score. If the data points produced by different sparse patterns are monotonically increasing, then it suggests that higher $Q^p$ will lead to higher accuracy, and $Q^p$ is a good metric for comparing different sparse patterns. The results are shown in Figure 12. It demonstrates that higher F1 is achieved with higher $Q^p$, so our $Q^p$ is a good metric for comparing different sparse patterns.
> >
> > **(5) How was the fixed sparsity pattern selected in figure 10?**
> >
> > **Response**: The fixed sparsity pattern in the original Figure 10 (now Figure 11) is the random selection under the Bernoulli distribution. As shown in Figure 17 of the revised paper, the important entries scatter all over the attention weight matrix. Different inputs and heads also have different distributions. So the fixed sparsity is equivalent to randomly sampling under Bernoulli distribution without finetuning. Theoretically, just like the no-free lunch theorem, any two fixed sparse patterns are equivalent when their performance is averaged across all possible problems. A pattern that is heuristically better than random sampling at a specific task can also be worse than random sampling on some other tasks.
> >
> > **(6) Could you please elaborate on the sentence "On the other hand, we have a theoretical value of $\sigma\approx 1$ and $p\ge 1$ based on the observation that the edges with higher magnitude are more influential." Specifically why $\sigma \approx 1$ if higher magnitude edges are more influential?**
> >
> > **Response**: The original expression is misleading. “higher magnitude edges are more influential” is used to explain why $p\ge 1$. $\sigma\approx 1$ basically comes its definition $qk^T/\sqrt{d}$, which is two normal distributed matrix multiplied together and normalized by dividing $\sqrt{d}$. In reality, we find that the $\sigma$ is usually greater and close to 1.

---

> > > ### Author Response · Authors · 2021-11-19
> > > **Response to Reviewer zwV3 (Part 3)**
> > >
> > > **Minor issues:**
> > >  * Page 16, before eq. 17, Therefore, $L^p$  quality of Top-K fixed sparsity.
> > >  * References missing in the table 4
> > >
> > > **Response**: Thank you for pointing these out. We have fixed these issues in the revised paper.
> > >
> > > **Reference**:
> > >
> > > [1] Lym, Sangkug, et al. "DeLTA: GPU Performance Model for Deep Learning Applications with In-Depth Memory System Traffic Analysis." 2019 IEEE International Symposium on Performance Analysis of Systems and Software (ISPASS). IEEE, 2019.

---

### Official Review · Reviewer_V8Q5 · 2021-11-03

**Correctness:** 3
**Technical Novelty And Significance:** 2
**Empirical Novelty And Significance:** 2
**Recommendation:** 5
**Confidence:** 4

**Main Review:**

Strength: A new idea of applying structured sparse patterns to approximate self-attention in Transformer.

Weakness:
1. Approximate self-attention analysis. Why quality of Attention lottery is a good indication of softmax matrix approximation $A$
, instead of the error between true softmax matrix $A$ and approximate softmax matrix $\tilde{A}$? Since $A$ is a softmax matrix and $A$ only has a 1 entry for each row, $Q^p = 1$ is not meaningful that approximate $A$ is the worst approximation when $m \odot A$ does not pick that entry.  The authors should give a more direct approximation analysis, such as $||A - \tilde{A}||_F$ or $||A - \tilde{A}||_F/||A||_F$, and present comparison with other baselines, such as longformer.

2. Efficiency analysis. Eq. 4 shows an upper bound of speedup. Why $s < 4.5\%$, leads to speedup $> 1$? Eq. 5 shows an upper bound. Why Eq. 6 shows an equality? How the comparison with Top-K sparsity presents an upper bound in Eq. 7 and Eq. 8 presents an approximation equality?

3. LRA benchmark. LRA benchmark has 5 tasks. Why only 3 tasks are used for comparison? Recommend to include state-of-the-art efficient self-attention methods on LRA, such as [1], [2], and [3]?

4. Speedup experiment. The authors only compare the self-attention part, which does not necessarily represent the speedup of the proposed method for Transformer (forward and backward). The authors should report the running time and memory consumption for a vanilla Transformer with respect to different sequence lengths and compare with other baselines as in [4]. How about the speedup on commonly used GPU, e.g., 2080Ti/V100s?

[1] Nyströmformer: A Nyström-Based Algorithm for Approximating Self-Attention

[2] SOFT: Softmax-free Transformer with Linear Complexity

[3] H-Transformer-1D: Fast One-Dimensional Hierarchical Attention for Sequences

[4] You Only Sample (Almost) Once: Linear Cost Self-Attention Via Bernoulli Sampling

**Summary Of The Paper:**

This paper presents DFSSATTEN, a fine-grained structured sparse attention mechanism.
Experiments show that the proposed method achieves speedups in A100 GPU.

**Summary Of The Review:**

This paper presents an efficient self-attention mechanism and achieves speedup. But the analysis and experiments have some issues (see weakness). I will give a borderline due to those concerns and change it accordingly based on rebuttal.

---

> ### Author Response · Authors · 2021-11-19
> **Response to Reviewer V8Q5 (Part 1)**
>
> Thank you for your constructive feedback. We have carefully revised our paper and the responses are listed as follows.
>
> **(1) Approximate self-attention analysis. Why is the quality of the Attention lottery a good indication of softmax matrix approximation $A$, instead of the error between true softmax matrix $A$ and approximate softmax matrix $\tilde{A}$?**
>
> **Response**: The quality of the Attention lottery in Definition 4.1 is a generalized direct approximation analysis. It can be easily transformed to the l2-norm of the errors in each row by having $p=2$ as follows:
> $$
> 1 - Q^p = \frac{1}{n}\sum_{j=1}^n\frac{\sum_{i=1}^n A_{j,i}^2 - (m_{j,i}\odot A_{j,i})^2}{\sum_{i=1}^n A_{j,i}^2}
> $$
> For each i, j, since $m$ is a binary mask, we have
> $$
> A_{j,i}^2 - (m_{j,i}\odot A_{j,i})^2 = (A_{j,i} - (m_{j,i}\odot A_{j,i}))^2
> $$
> The above equation can be simply proved by substituting $m=1$ and $m=0$ to both sides of the equality. Therefore, we have
> $$
> 1-Q^p = \frac{1}{n}\sum_{j=1}^n\frac{||A_{j,}-(m_{j,}\odot A_{j,}) ||^2}{||A_{j,}||^2}.
> $$
> We use the row-wise Lp norm rather than the F-norm of the whole matrix because A is usually normalized and used row-wisely.
>
> We do not fix $p$ to specific values like 1 or 2 because we observe that the importance of the largest entries in the attention weight matrix varies in different tasks. A large $p$ can model the tasks that depend more on the larger entries and vice versa. In the theoretical analysis, we also prove that our conclusions hold for any $p>1$ rather than a specific $p$.
>
> To support this point, we plot the $Q^p\sim$F1 and $||A-(m\odot A)||_F^2//||A||_F^2\sim$F1 obtained from different sparsity patterns in BERT-large, SQuAD v1.1 in Figure 12. The $Q^p\sim$F1 is almost monotonically increasing, which suggests that a higher $Q^p$ leads to a higher F1 score. Oppositely, $||A-(m\odot A)||_F^2//||A||_F^2$ cannot explain why our 1:2 Sparsity has a higher F1 score than some points from the fixed sparsity, even though it has a larger approximation error in terms of F-norm.
>
> **(2) Since $A$ is a softmax matrix and $A$ only has a 1 entry for each row, $Q^p=1$ is not meaningful; that approximate $A$ is the worst approximation when $m\odot A$ does not pick that entry.**
>
> **Response**: While $A$ is a softmax, it usually has multiple rather than one entry with relatively high magnitude in each row. For instance, if we only keep the maximum entry in each row of $A$, the F1 score will drop from 93.2 to 56.0. Oppositely, even if the largest entry is not picked by mask $m$, as long as other large entries are picked, there is still a good chance to preserve the accuracy. We further plot the attention weight magnitude in Figure 17 of the revised paper to show this point.
>
> **(3) The authors should give a more direct approximation analysis, such as $||A-\tilde{A}||_F$ or $||A-\tilde{A}||_F/||A||_F$, and present comparison with other baselines, such as longformer.**
>
> **Response**: Thank you for your suggestions. Our theoretical analysis compares three types of baselines: Sparse attention with top-K sparsity like ElSA [7], Sparse attention with fixed sparsity like Longformer, and low-rank/kernel-based attention like Performer.
>
> The longformer you mentioned is a special case of fixed sparsity. Our theoretical results have shown that the fixed sparse mask is less effective than our 1:2 and 2:4 sparsity. Actually, the attention distribution bias used in Longformer or BigBird does not naturally exist in the model pretrained with full attention. As shown in Figure 17 of the revised paper, the important attention weights scatter all over the attention weight matrix. Therefore, the fixed mask used in models like Longformer or Bigbird is more like the inductive bias in CNN, which requires costly pretraining to force the distribution of the attention weights to their biased sparse pattern. So they are unfriendly for lightweight finetuning.
>
> The comparison with the Performer [5], a representative kernel-based efficient transformer, is in Appendix A.5 of the original submission.  As our $L^p$-Quality is primarily designed for measuring how the sparse pattern captures the important entries, it is not suitable for the kernel/low-rank-based methods. Hence, the comparison with Performer[5] is based on the MSE originally used in [5]. We observe that the Performer has high MSE for large magnitude attention weights, which is the price of preventing the MSE of small magnitude entries from going to infinity. Nevertheless, the less accurate approximation of the large magnitude entries makes it less friendly to the lightweight finetuning. On the other hand, our method has low MSE for large magnitude entries as they are more likely to be chosen. The MSE for small magnitude entries is also bounded by their magnitude. This makes our method more friendly to lightweight finetuning.

---

> > ### Author Response · Authors · 2021-11-19
> > **Response to Reviewer V8Q5 (Part 2)**
> >
> > **(4) Efficiency analysis. Eq. 4 shows an upper bound of speedup. Why $s<4.5$, leads to speedup $>1$? Eq. 5 shows an upper bound. Why Eq. 6 shows equality? How the comparison with Top-K sparsity presents an upper bound in Eq. 7 and Eq. 8 presents approximation equality?**
> >
> > **Response**: Thank you for pointing out these typos. The original statement about Eq. 4 is a little bit misleading. When $s>4.5%$, Eq .4 shows that Speedup $< 1$. So having $s<4.5$ is a necessary and insufficient condition for Speedup $>1$. Eq.5 and Eq.6 should also be equalities.
> >
> > The Speedup of top-K given in Eq. 4 is formulated without considering the overhead of identifying the location of important entries and encoding them to compressed sparse row format. So the actual speedup is even lower, and lower density is required for speedup. That’s why Eq.7 is an upper bound. Both fixed sparsity and our 1:2/2:4 sparsity do not involve such prediction or encoding overhead. So the speedup of both methods can be computed explicitly. That’s why Eq.8 represents equality.
> >
> > We have fixed these issues in the revised paper.

---

> > > ### Author Response · Authors · 2021-11-19
> > > **Response to Reviewer V8Q5 (Part 3)**
> > >
> > > **(5) LRA benchmark. LRA benchmark has 5 tasks. Why only 3 tasks are used for comparison? Recommend to include state-of-the-art efficient self-attention methods on LRA, such as [1], [2], and [3]?**
> > >
> > > **Response**: We originally included 3 tasks to cover the sequence lengths 1024, 2048, and 4096 to demonstrate the effectiveness of our method under long sequences. In the revised paper, we also include the results from ListOps. For Pathfinder (1K), we tried both the configuration from the LRA paper [6] and Nyströmformer [1] but we can not reproduce the results of the baseline dense transformer. This issue is also reported in [2]. Therefore, we omit this task from table 4.
> > >
> > > Following your suggestion, we add [1] to the speedup comparison in Section 5.2 and Appendix A.6 of the revised paper. Our method achieves higher speedup than theirs when the sequence length $\le 2048$. The throughput of [2] is reported to be lower than [1], and we did not observe any speedup with [3] over the dense transformer under sequence length 4096. So we omit [2] and [3] for speedup comparison.
> > >
> > > In terms of accuracy on LRA, our method can be combined with existing efficient transformers to achieve higher efficiency with the same or higher accuracy. In Appendix A.7 of the revised paper, we combine DFSSATTEN with Nystromformer [1] you mentioned and measure the accuracy of Image in LRA. The combined model has 0.4 and 1 higher accuracy than the standard Nystromformer under 1:2 and 2:4 sparsity. It also has lower latency and memory footprint.
> > >
> > > When compared alone, we found it is difficult to include [1], [2], and [3] to our Table 4 for a fair comparison in terms of LRA scores during the rebuttal. [1] and [2] use a customized LRA benchmark in pytorch rather than the official one in Jax. Due to some configuration differences, their accuracy on Retrieval (4K) is 21.89 higher than the original results in [2]. So it is unfair to directly include their accuracy in Table 4. On the other hand, we do not have enough time to reimplement these two methods in JAX and measure their accuracy during the rebuttal. [3] only report the model size and scores on LRA, and our evaluation does not show speedup over the dense baseline. So even though it achieves good scores in all tasks, this method focuses on improving accuracy with hierarchical inductive bias and reducing memory footprint rather than achieving speedup.
> > >
> > > Our method is more like a complementary of existing efficient transformer models rather than a challenger. It has several merits that are missing in existing methods.
> > >
> > > First, as shown in Figure 13 of the revised paper, our method is the only one that achieves end-to-end inference speedup in all the sequence lengths. Although [1] also reported speedup overall sequence lengths in their paper, after studying their code, we find that this speedup comes from the sub-optimal baseline implementation. After applying JIT compilation in both baseline and Nystrom, we found that it requires sequence length $\ge$ 2048 to achieve speedup, and our method performs better than it does when the sequence length $\le 2048$. In detail, dividing the attention score matrix with $\sqrt{d}$ and adding padding mask (line 15, 16) in (https://github.com/mlpen/Nystromformer/blob/main/code/attention.py) are not fused into either the softmax kernel or the $QK^T$ kernel. This optimization is simple and can be done even with a compiler. Although this kind of optimization is also missing in the implementation of their own method, the impact is different: there are two additional $O(n^2)$ operators in the dense transformer, whereas Nystrom only has some additional $O(nd)$ operators that are much smaller.
> > >
> > > Second, it supports lightweight finetuning from pre-trained dense models. This enables us to directly leverage existing pretrained models like BERT and GPT-3 that take millions of dollars to train. For instance, it only takes half an hour for it to finetune the pre-trained BERT-large checkpoint to downstream tasks like SQuAD v1.1. Our Table 2 & 3 also shows that directly applying our DFSSATTEN without finetuning still yields good performance. Oppositely, existing efficient transformers require training from scratch or retraining from the dense pretrained model. For instance, the Nystromformer takes 0.5M steps (14 GPU days on V100) to train a BERT-base model from scratch or 0.25M (7 GPU days on V100) to train from the pre-trained BERT-base. Not to mention the 3x larger BERT-large takes more resources and carbon emission to train.

---

> > > > ### Author Response · Authors · 2021-11-19
> > > > **Response to Reviewer V8Q5 (Part 4)**
> > > >
> > > > Third, our method introduces fewer inductive biases than other methods. Previous studies require extremely high sparsity or low rank to compensate for the overhead they introduced. So when the task does not match the bias, the model works poorly compared with the dense transformer. For instance, as shown in Table 4, the low rank/kernel-based transformers perform poorly on ListOps and Retrieval tasks. Oppositely, our method only requires 50% sparsity, so it still has enough volume to meet the requirement for different tasks. Therefore, we achieve comparable accuracy with the dense transformer in all tasks.
> > > >
> > > > **(6) Speedup experiment. The authors only compare the self-attention part, which does not necessarily represent the speedup of the proposed method for Transformer (forward and backward). The authors should report the running time and memory consumption for a vanilla Transformer with respect to different sequence lengths and compare with other baselines as in [4]. How about the speedup on commonly used GPU, e.g., 2080Ti/V100s?**
> > > >
> > > > **Response**: Our method targets efficient inference. Following your suggestion, we report the end-to-end speedup and memory consumption of the vanilla transformer, our method, and other baselines in the Appendix A.6 of the revised paper following [4]. Our method achieves 1.08$\sim$1.47x end-to-end speedup and 1.41$\sim$1.82x memory reduction across various configurations. It beats most of the baselines when the sequence length $\le 2048$, which justifies that our method is more effective under moderate and short sequence lengths. It is the only method that achieves end-to-end speedup in all the measured configurations. For longer sequences, as discussed in Appendix A.7, our method can be combined with existing efficient transformers for higher speedup. Besides, as mentioned before, our method has other merits such as it supports lightweight finetuning.
> > > >
> > > > Our method cannot bring speedup on commonly used GPUs like 2080 Ti or V100s, as it requires the sparse tensor core on Ampere architecture. Nevertheless, just like the 1st generation tensor core on V100 spread to all post-volta GPUs within 3 years, we believe the sparse tensor core will also be a common feature in the forthcoming GPU models in the foreseeable future. Then our method will have a higher impact.
> > > >
> > > > **References**
> > > >
> > > > [1] Xiong, Yunyang, et al. "Nyströmformer: A Nyström-based Algorithm for Approximating Self-Attention." Proceedings of the AAAI Conference on Artificial Intelligence. Vol. 35. No. 16. 2021.
> > > >
> > > > [2] Lu, Jiachen, et al. "SOFT: Softmax-free Transformer with Linear Complexity." arXiv preprint arXiv:2110.11945 (2021).
> > > >
> > > > [3] Zhu, Zhenhai, and Radu Soricut. "H-Transformer-1D: Fast One-Dimensional Hierarchical Attention for Sequences." arXiv preprint arXiv:2107.11906 (2021).
> > > >
> > > > [4] Zeng, Zhanpeng, et al. "You only sample (almost) once: Linear cost self-attention via bernoulli sampling." International Conference on Machine Learning. PMLR, 2021.
> > > >
> > > > [5] Choromanski, Krzysztof Marcin, et al. "Rethinking Attention with Performers." International Conference on Learning Representations. 2020.
> > > >
> > > > [6] Tay, Yi, et al. "Long Range Arena: A Benchmark for Efficient Transformers." International Conference on Learning Representations. 2020.
> > > >
> > > > [7] Ham, Tae Jun, et al. "ELSA: Hardware-Software Co-design for Efficient, Lightweight Self-Attention Mechanism in Neural Networks." 2021 ACM/IEEE 48th Annual International Symposium on Computer Architecture (ISCA). IEEE, 2021.

---

### Official Review · Reviewer_cVkD · 2021-11-04

**Correctness:** 3
**Technical Novelty And Significance:** 3
**Empirical Novelty And Significance:** 2
**Recommendation:** 5
**Confidence:** 3

**Main Review:**

Strengths:

This paper offers a simple method with clean code implementation to leverage sparse tensor cores for attention speedup. It demonstrates that the 50% sparsity pattern can retain model performance very well, and it yields 1.27~1.89x practical speedup over the vanilla attention across different sequence lengths.

Weaknesses:

* The speed analysis is narrowed down to the attention alone, which is not *practical*.
* Comparison across different models might not be fair.
* The proposed method is very specific to the sparse pattern offered by NVIDIA A100, which is not generalizable.


Detailed comments:
1)	Since the major goal of this paper is to achieve “practical” speedup over baselines, showing speedups over the attention component alone is far from enough and not *practical* at all. As a user, I would care about whether the proposed model could make the overall training faster or could allow larger-batch training so that reduce the gradient accumulation steps. Unfortunately, neither is analyzed. Based on my own experiments, attention computation, i.e. QK, softmax, and AV used in the paper, only takes a small proportion of the whole Transformer. To prove the significance of DFSSATTEN, more analysis should be given.
a)	Particularly, I would like to see further analysis on the speedup of the whole Transformer with respect to sequence length, model dimension and model depth.
2)	The proposed method is based on new sparse operations supported by the latest NVIDIA GPUs, or A100. The fact that A100 is very expensive, and many institutes can’t afford it at this moment largely devalues this paper. How does the model perform on other old GPUs, like 1080 and 2080 GPUs? Would we get slower running speed? Also, the proposed attention method doesn’t solve the quadratic attention complexity issue. So, the generalization ability of DFSSATTEN is highly doubtful.
3)	The speedup given by DFSSATTEN is mostly from the low-level optimization and hardware support. However, many existing algorithms like Performer and Reformer are often not deeply optimized. So directly comparing with them might be unfair. Besides, in introduction, the authors argue that these methods are usually trained from scratch. This is actually a very significant direction since pretraining becomes very popular and important nowadays. Then a question is whether DFSSATTEN supports training from scratch. It would be great if DFSSATTEN could accelerates the training from scratch, but this is not studied in the paper.


**Summary Of The Paper:**

This paper aims at improving the computational efficiency of the attention mechanism by leveraging the specific sparse pattern supported by sparse tensor cores of NVIDIA A100, with a particular focus on delivering practical running speedup. To achieve this, the authors proposed DFSSATTEN, which shows both theoretical and empirical advantage in terms of performance and speedup compared to various baselines. In particular, DFSSATTEN yields 1.27~1.89x speedup over the vanilla attention network across different sequence lengths.

**Summary Of The Review:**

In summary, this paper provides a solution based on sparse tensor cores (specific to NIVIDIA A100) to optimize and accelerate attention operations. Although it yields good performance and nice attention speedup, its practical value on Transformer efficiency is unclear and its generalization to other devices is also unclear (at least, we shouldn’t get slow running on other GPUs or CPUs).

---

> ### Author Response · Authors · 2021-11-19
> **Response to Reviewer cVkD (Part 1)**
>
> We sincerely appreciate your valuable comments. We have carefully revised the paper and the responses are listed as follows.
>
> **(1) The speed analysis is narrowed down to the attention alone, which is not practical. Since the major goal of this paper is to achieve “practical” speedup over baselines, showing speedups over the attention component alone is far from enough and not practical at all. As a user, I would care about whether the proposed model could make the overall training faster or could allow larger-batch training so that it reduces the gradient accumulation steps. Unfortunately, neither is analyzed. Based on my own experiments, attention computation, i.e. QK, softmax, and AV used in the paper, only takes a small proportion of the whole Transformer. To prove the significance of DFSSATTEN, more analysis should be given. a) Particularly, I would like to see further analysis on the speedup of the whole Transformer with respect to sequence length, model dimension, and model depth.**
>
> **Response**: Thank you for your suggestions! Following your questions, we add additional analysis on end-to-end inference speedup and memory footprint reduction in the new Appendix A.6. Our method achieves 1.08$\sim$1.52x speedup and 1.41$\sim$1.82x memory reduction in terms of the whole Transformer under different sequence lengths, model dimensions, etc.
>
> Our method targets accelerating the attention mechanism in the inference. Attention is the key to the stunning performance of transformers. It can also be the performance bottleneck during inference under all different sequence scenarios.
>
> Figure 14 of the revised paper shows that the attention alone contributes over 60% of the total inference time when the sequence length $\ge 2048$. Accelerating this part gives us over 1.4x end-to-end speedup.
>
> Under shorter sequence lengths like 512 and 1024, the attention mechanism still could be the bottleneck during inference. This is because there are tons of techniques that compress the non-attention parts in the literature even before transformers are invented, including pruning, quantization, and distillation. For example, [3] shows 2:4 structured sparsity in weights brings 1.3~1.6x speedup to the linear layers in BERT-Large. As these methods cannot be applied to the attention mechanism, they will make the remaining attention part the new bottleneck under short sequence lengths.
>
> **(2) The proposed method is very specific to the sparse pattern offered by NVIDIA A100, which is not generalizable. The proposed method is based on new sparse operations supported by the latest NVIDIA GPUs, or A100. The fact that A100 is very expensive, and many institutes can’t afford it at this moment largely devalues this paper. How does the model perform on other old GPUs, like 1080 and 2080 GPUs? Would we get a slower running speed?**
>
> **Response**: Indeed, our method requires the latest sparse tensor core in A100 to achieve speedup. However, this does not weaken the generalization ability of our method.
>
> First, the N:M  fine-grained structured sparsity is becoming a popular trend in both academia and industry. In academia, studies like [5] (ICLR 2021) and [6](NeurIPS 2021) all focus on accelerating deep learning workloads with N:M fine-grained sparsity. Same with ours, they also require the sparse tensor core on A100 to achieve speedup. While these papers focus on static offline pruning, we are the first to adopt it in the dynamic pruning scenario which is more challenging. Besides, our method also supports general N:M fine-grained structured sparsity given the hardware support. In industry, the sparse tensor core is adopted in all NVIDIA’s Ampere GPUs including the cheaper RTX 30 series. It is even used in other hardware besides GPUs. For example, [1] introduce the Density-Bound Block (DBB). DBB constrains the number of nonzeros in a small block, e.g. NNZ <= 4 in an 8 x 1 vector, which is very similar to the sparse tensor core.
>
> Besides, A100 has a bag of new features that accelerate dense matrix multiplication. These features cannot be fully leveraged by other operations. Therefore, it is more challenging to beat the dense attention mechanism on A100 GPU. Other efficient transformers that perform well on previous-generation GPUs may be less effective on Ampere. This makes our method more meaningful.

---

> > ### Author Response · Authors · 2021-11-19
> > **Response to Reviewer cVkD (Part 2)**
> >
> > **(3) Also, the proposed attention method doesn’t solve the quadratic attention complexity issue. So, the generalization ability of DFSSATTEN is highly doubtful.**
> >
> > **Response**: Our method is a good complementary of existing efficient transformers that solve quadratic complexity. It can be directly combined with methods like Nystromformer, BigBird, or Linformer. In the Appendix A.7 of the revised paper, we combine our DFSSATTEN with the Nystromformer [2] and evaluate its accuracy on Image (1K) of LRA benchmark. With a lightweight finetuning, our method achieves 0.4 and 1 higher accuracy over the standard Nystromformer. Moreover, it further reduces the compute and memory complexity.
> >
> > Besides, Our method is a more practical solution to accelerate existing transformer models, even though it alone does not solve the quadratic attention problem under asymptotic complexity. Figure 13 in the revised paper shows that our method achieves higher speedup than existing efficient transformers when the sequence length $\le 2048$. Notably, most transformers nowadays are still under moderate or short sequence length (384 for BERT and 2048 for GPT-3). Besides, our method also supports lightweight finetuning while others don’t.
> >
> > **(4) Comparison across different models might not be fair. The speedup given by DFSSATTEN is mostly from the low-level optimization and hardware support. However, many existing algorithms like Performer and Reformer are often not deeply optimized. So directly comparing them might be unfair.**
> >
> > **Response**: During the evaluation, we also process the baselines with the PyTorch just-in-time (JIT) compilation for a fair comparison. The PyTorch JIT is a deep learning compiler embedded in PyTorch that can leverage many low-level optimizations like kernel fusion. This is a conventional way for fair comparison adopted widely by previous studies. We are willing to add additional comparisons if you have any suggestions.
> >
> > Leveraging new hardware on A100 does not devalue the algorithm side contribution of our paper. The 2:4 sparse tensor core is initially introduced to support static pruning of weights in deep neural networks, and our paper is the first one that demonstrates it works well on dynamic pruning. To the best of our knowledge, it is also the first method that dynamically generates and encodes fine-grained sparse matrices under a compact format in the deep learning software stack.
> >
> > Besides, the low-level optimization in DFSSATTEN is only a sub-operator optimization. Oppositely, existing models like Performer require whole-model optimization. The latter one is far more complex.
> >
> > In detail, we only require adding a single epilogue that prunes the output of the dense matrix multiplication kernel (less than 300 lines of C++ code in “src/cuda/utils/gemmv2/meta_util.h”). The remaining part of our codebase is just reimplementing some basic BLAS kernels and wrappers.
> >
> > On the other hand, when optimizing models like Performer, it takes tremendous engineering effort to explore different computation graph partitions, implement fused kernels composed of different types of operators in each subgraph, and tune the tiling sizes of these fused kernels. For instance, Turbotransformer [7] has thousands of lines of C++ codes (https://github.com/Tencent/TurboTransformers/tree/master/turbo_transformers/layers/kernels) that implement customized kernels to accelerate a naive transformer. Not to mention that efficient transformers have much more complex compute graphs to optimize.
> >
> > Our kernel also has high reusability. As we mentioned before, our method is the first one that can prune and encode sparse matrices on the fly in the deep learning software stack. Therefore, it can be added to libraries like CUTLASS and reused in other models besides the attention mechanism. Oppositely, the deeply optimized kernels for Performer and Reformer are usually specialized for a single model and single configuration, which limits the reusability.
> >
> > Last but not least, our method can be further combined with existing efficient transformers.

---

> > > ### Author Response · Authors · 2021-11-19
> > > **Response to Reviewer cVkD (Part 3)**
> > >
> > > **(5) Besides, in the introduction, the authors argue that these methods are usually trained from scratch. This is actually a very significant direction since pre-training has become very popular and important nowadays. Then a question is whether DFSSATTEN supports training from scratch. It would be great if DFSSATTEN could accelerate the training from scratch, but this is not studied in the paper.**
> > >
> > > **Response**: Our method targets efficient inference, which is another significant direction besides training from scratch. This is because after the model is trained, it can be duplicated to many devices and serve for many years.
> > >
> > > Although our method alone does not accelerate the training from scratch. As the Nystromformer example in Appendix A.7 shows, our DFSSATTEN can further boost the accuracy and efficiency of other efficient transformers with lightweight finetuning.
> > >
> > > Moreover, Our method only requires low-cost fine-tuning to achieve comparable accuracy with the full attention mechanism, this is important merit missing in previous efficient transformers. The reason is that training a transformer model is expensive (GPT-3 costs $12 Million in a single training round). Even with efficient transformers like Nyströmformer [2], it still takes more than 7 GPU days on V100 to train a BERT-base from the pretrained dense BERT model. Not to mention the 3x larger BERT-large used in our paper and the 1000x larger GPT-3. Many institutes cannot afford this scale of pretraining. Besides, there is also a bag of hyper-parameters to be tuned to achieve the best performance. So a single training round may not be enough. Therefore, the industry hesitates to adopt the efficient transformer models due to the cost. Oppositely, our method can simply take a pretrained model like BERT-large, finetuning it on downstream tasks for less than one hour. This neglectable cost brings good savings for both time and memory space.
> > >
> > > Besides, It is very challenging to find a good approximation of the full attention mechanism that achieves speedup on GPUs without significantly changing the model. We believe this is one major reason for using pre-training in previous studies. Our method addresses this challenge, which makes it a good complementary for existing efficient transformer models even though it does not accelerate training from scratch.
> > >
> > > **References**
> > >
> > > [1] Liu, Z. G., Whatmough, P. N., & Mattina, M. (2020). Systolic tensor array: An efficient structured-sparse gemm accelerator for mobile cnn inference. IEEE Computer Architecture Letters, 19(1), 34-37.
> > >
> > > [2] Xiong, Yunyang, et al. "Nyströmformer: A Nyström-based Algorithm for Approximating
> > >
> > > [3] https://developer.nvidia.com/blog/exploiting-ampere-structured-sparsity-with-cusparselt/
> > >
> > > [4] Zhu, Maohua, et al. "Sparse tensor core: Algorithm and hardware co-design for vector-wise sparse neural networks on modern gpus." Proceedings of the 52nd Annual IEEE/ACM International Symposium on Microarchitecture. 2019.
> > >
> > > [5] Zhou, Aojun, et al. "Learning N: M Fine-grained Structured Sparse Neural Networks From Scratch." International Conference on Learning Representations. 2020.
> > >
> > > [6] Pool, Jeff, and Chong Yu. "Channel Permutations for N: M Sparsity." Thirty-Fifth Conference on Neural Information Processing Systems. 2021.
> > >
> > > [7] Fang, Jiarui, et al. "TurboTransformers: an efficient GPU serving system for transformer models." Proceedings of the 26th ACM SIGPLAN Symposium on Principles and Practice of Parallel Programming. 2021.
> > >
> > > [8] Beltagy, Iz, Matthew E. Peters, and Arman Cohan. "Longformer: The long-document transformer." arXiv preprint arXiv:2004.05150 (2020).

---

### Decision · Program_Chairs · 2022-01-20

**Decision:**

Reject

**Comment:**

This paper presents a package for "Dynamic Fine-grained Structured Sparse Attention Mechanism" (DFSSATTEN), which aims to improve the computational efficiency of attention mechanisms by leveraging the specific sparse pattern supported by sparse tensor cores of NVIDIA A100. DFSSATTEN shows theoretical and empirical advantage in terms of performance and speedup compared to various baselines, with 1.27~1.89x speedup over the vanilla attention network across different sequence lengths.

Reviewers praised the simplicity of the method and the clean code implementation. Speeding up attention mechanisms is an important problem is leveraging sparse tensor cores for attention speedup is a sensible idea. The practical speedups are significant (1.27~1.89x over the vanilla attention across different sequence lengths). However, they also pointed out some weaknesses: the fact that the proposed method is very specific to the particular sparse pattern offered by NVIDIA A100, and not easily generalizable to other future hardware; the fact that the method focuses on inference acceleration and not training from scratch (not completely clear in the paper), which limits its scope; and the fact that the method still has O(N^2) complexity (it still requires the computation of QK^T, which has quadratic memory and computation cost), and therefore it does not really address the quadratic bottleneck of transformers, unlike other existing work in efficient transformers for long sequences.

I tend to agree with the reviewers and, even though the package can be potentially useful to other researchers, the scope seems limited and the paper seems a bit thin to deserve publication at ICLR.

Other comments and suggestions:
- When talking about linear transformers, you should cite [1], which predates Performers
- It is not clear to me why 1:2 and 2:4 are called "fine-grained *structured* sparsity"
- Citations for the systems in Tab 4 are missing
- When comparing to other methods, it would be include to include their Pareto curves since those methods have tradeoffs in terms of sparsity / approximation error (or downstream accuracy).

[1] Transformers are RNNs: Fast Autoregressive Transformers with Linear Attention. Angelos Katharopoulos, Apoorv Vyas, Nikolaos Pappas, François Fleuret (https://arxiv.org/abs/2006.16236)